# Foraging in a non-foraging task: Fitness maximization explains human risk preference dynamics under changing environment

**Yasuhiro Mochizuki**[1]*, **Norihiro Harasawa**[2☙], **Mayank Aggarwal**[2☙], **Chong Chen**[3], **Haruaki Fukuda**[4]

**1** Center for Data Science, Waseda University, Shinjuku-ku, Tokyo, Japan, **2** RIKEN Center for Brain Science, Wako City, Saitama, Japan, **3** Division of Neuropsychiatry, Department of Neuroscience, Yamaguchi University Graduate School of Medicine, Ube, Yamaguchi, Japan, **4** Graduate School of Business Administration, Hitotsubashi University, Kunitachi, Tokyo, Japan

☙ These authors contributed equally to this work.
* yasuhiro.mochizuki87@gmail.com

**Data Availability Statement:** All data are available at https://github.com/yazmochi/forageNonforage.

**Funding:** The author(s) received no specific funding for this work.

## Abstract

Changes in risk preference have been reported when making a series of independent risky choices or non-foraging economic decisions. Behavioral economics has put forward various explanations for specific changes in risk preference in non-foraging tasks, but a consensus regarding the general principle underlying these effects has not been reached. In contrast, recent studies have investigated human economic risky choices using tasks adapted from foraging theory, which require consideration of past choices and future opportunities to make optimal decisions. In these foraging tasks, human economic risky choices are explained by the ethological principle of fitness maximization, which naturally leads to dynamic risk preference. Here, we conducted two online experiments to investigate whether the principle of fitness maximization can explain risk preference dynamics in a non-foraging task. Participants were asked to make a series of independent risky economic decisions while the environmental richness changed. We found that participants' risk preferences were influenced by the current and past environments, making them more risk-averse during and after the rich environment compared to the poor environment. These changes in risk preference align with fitness maximization. Our findings suggest that the ethological principle of fitness maximization might serve as a generalizable principle for explaining dynamic preferences, including risk preference, in human economic decision-making.

## Author summary

Decision-making involving probabilistic outcomes (i.e., under risk) is an integral part of our daily lives. Empirical studies have shown that risky behavior often deviates from standard economic theory, and peoples' risk preferences change depending on their psychological and physiological states, and the context of the decision. While previous studies have developed empirical mathematical models to provide mechanistic explanations of these effects, there is no unifying principle that explains why human risky decision-making is tuned this way. Here, we document one such context effect and suggest that this

**Competing interests:** The authors have declared that no competing interests exist.

effect can be explained by the ethological principle of fitness maximization. In our study, participants made sequential independent risky decisions in different environments. We found that rich environments in which it was easy to get large rewards increased participants' risk-aversiveness both during and after experiencing them, and experiencing poor environments increased risk taking behavior. The foregoing modulations of risk-aversiveness are predicted if participants make decisions to satisfy some internal threshold for minimum reward gain, akin to reaching a minimum threshold for survival, rather than to maximize reward gain. Our results suggest that a better understanding of human economic behavior may be achieved under the principle of fitness maximization.

## Introduction

How humans choose between options with probabilistic outcomes is a major question in economics and psychology [1,2]. This problem has been formally investigated by studying risky choice. In this context, "risk" specifically refers to the variance in the outcome of an option, as defined in economics. The standard models of risky choice assume that individuals make decisions to maximize utility [3,4], where the utility of an option is determined by both the option's properties and the risk preference of the decision-maker. In classical theories of risky choice, such as expected utility theory [3,4], modern portfolio theory [5] and prospect theory [6], an individual's risk-preference is assumed to be a stable trait [7].

However, the framing of the decision problem [8,9], the history of prior outcomes [10,11], an individual's emotional state [12–16], hormone levels [17–19], observation of other's choices [20], and aging [21] can alter the risk preference of a decision-maker. The foregoing observations show that the constancy of risk preference does not always hold [22]. Numerous studies have endeavored to elucidate the mechanistic underpinnings of changes in risk preference by employing diverse computational models with intricate utility functions tailored to specific contexts [23–28]. Notably, models with divisive normalization [24,27,29,30], reference-point centering [26,31,32], and range adaptation [26,33,32] have prominently featured in this exploration. However, while these studies proficiently elucidate the mechanisms through which dynamic modulation of risk preference arises as a consequence of learning-based utility modulation, they lack a comprehensive articulation of the adaptive significance that underlies the specific structure of the utility function and its evolution over time.

In contrast to these studies, ethological theories assume that animals' risky choices are guided by the maximization of long-term fitness rather than utility, which naturally leads to dynamic risk preference [34,35]. For example, the risk-sensitive foraging theory predicts that animals should be risk-prone in environments in which the mean rate of food gain is less than the animals' energetic needs, and vice-versa [34,36–38]. In accordance with the foregoing prediction, Caraco et al. observed that birds' preference between two food patches, which delivered the same amount of food on average but differed in risk, changed depending on their energy needs [39]. Under higher energy needs (e.g., cooler temperature or greater starvation), birds preferred the food patch with higher risk. Stephens argued that the foregoing shift in risk preference is inevitable if the average energy delivered from an environment is insufficient to support their energy needs [36]. Therefore, fitness maximization rather than utility maximization better describes the foregoing patch preference of birds.

Recent human studies investigating human economic risky choices in tasks adapted from foraging theory [40,41] have found that experimentally induced needs change human risk preference in accordance with risk-sensitive foraging theory [42–46]. Critically, such foraging

tasks differ from non-foraging tasks typically used to study human risky choice in that the optimality of the decision cannot be defined independently for each trial and depends on the outcome of past decisions and/or expected future opportunities. For example, Kolling *et al.* required participants to accumulate rewards to a target amount within a block of eight risky choices so that they could receive the accumulated reward for that block [45]. As a result, the participants' choices became more risk-prone with increasing need, as defined by the discrepancy between the target and accumulated reward divided by the number of remaining trials in the block. The forgoing shift in risk preference is in accordance with risk-sensitive foraging theory and suggests that fitness maximization rather than utility maximization better describes human economic choice in foraging situations.

To explore whether the observed shifts in human risk preference in foraging and non-foraging tasks can be understood under the same principle, here we investigated whether the risk-sensitive foraging theory and, thus, the principle of fitness maximization explains human risk preference dynamics in a non-foraging task. In our experiment, participants made independent risky choices in blocks of differing environmental richness. Classical economic theories assume that the optimality of choices depends only on the choice at hand and thus predicts no shifts in participants' risk preference across different environments. In contrast, the risk-sensitive foraging theory predicts that participants' risk preference in a given environment will be modulated by the mean rate of gain in the environment as well as the reward accumulated in the past. In particular, participants are predicted to be more risk-prone for the same choice options in poor environments where the mean rate of gain is lower than in rich environments. In addition, participants accumulate less reward in poor than in rich environments and are thus predicted to be more risk-prone after experiencing poor than after experiencing rich environments. Thus, the risk-sensitive foraging theory makes specific predictions about the effects of the current and past environments on the participants' risk preference.

The effect of the current and past environments on participants' risk preference predicted by the risk-sensitive foraging theory is intuitive in an affect-based explanation. An option with a reward probability of 50% seems more attractive in a poor environment with a typical reward probability of 40% compared to a rich environment with typical reward probabilities of 60%. Similarly, an option with a reward probability of 50% in the current environment seems more attractive when the environment is an improvement from a previously experienced poor environment with a typical reward probability of 40% rather than a deterioration from a previously experienced rich environment with a typical reward probability of 60%.

Our results were in accordance with the predictions of risk-sensitive foraging theory. First, we found that participants were consistently more risk-averse in the rich than in the poor environment for the same set of choice options. Second, prior exposure to the rich and poor environments made participants more and less risk-averse, respectively. Regression analysis revealed that, along with the expected value and the risk of an option, the participants' choices were consistently modulated by two environment-related regressors, as well as the accumulated reward. One environmental regressor captured the effect of the richness of the current environment, while the other captured the effect of the past environment on the participants' choices. Follow up simulations showed that the environment related regressors were critical in reproducing the participants' observed risk preference dynamics, but the accumulated reward regressor was not.

Taken together, our results show that both environmental richness and the order in which the environments are experienced affect human risk preference during economic decision-making in a non-foraging task, as predicted by the risk-sensitive foraging theory. Our findings suggest that humans make risky choices to maximize long-term fitness rather than utility, and

the observed shifts in risk preference in foraging and non-foraging tasks can be understood under the same principle of fitness maximization.

## Results

### Environmental richness affects participants' risk preferences

We conducted two online experiments in which participants chose whether to accept or reject risky gambles (Fig 1A) while experiencing environments of different richness (different reward probability distributions; Fig 1B and 1C). Three environments (poor, intermediate, and rich in the increasing order of mean reward probability) were used (Fig 1B). In Experiment 1, one group of participants experienced the poor environment followed by the rich environment (Poor-Rich or PR group), and the other group experienced the same environments in the reverse order (Rich-Poor or RP group; Fig 1C, left). In Experiment 2, one group experienced the poor environment, and the other group experienced the rich environment in-between two intermediate environments (Intermediate-Poor-Intermediate or IPI group and Intermediate-Rich-Intermediate or IRI group, respectively; Fig 1C, right). The rationale behind introducing Experiment 2 was to assess the consistency of our results across different experiments and to control for the influence of the environment that participants had encountered in previous trials. In both experiments, participants were not given instructions regarding the presence of distinct environments or the sequence in which they would encounter them.

On each trial, participants were presented with a risky gamble whose probability of success (reward probability) and the associated gain (reward magnitude) were explicitly shown (Fig 1A). Accepting the gamble led to a gain equal to the reward magnitude if successful and led to nothing otherwise. Rejecting the gamble led to a guaranteed gain of 10 points. The reward probability and magnitude were varied across trials. The proportion of gambles accepted increased significantly with increasing reward magnitude and reward probability (S1 Fig), showing that the participants understood and performed the task properly.

Fitness maximization predicts that participants' risk preference will be modulated by the environmental richness, such that the same option seems more attractive when experienced in the poor environment than when experienced in the rich environment. This predicts that participants will accept more gambles in the poor environment than in the rich environment. In contrast, utility maximization predicts that environmental richness will not have any effect on the participants' risk preference. To illustrate this disparity in predictions arising from fitness and utility maximizations, we conducted simulations of participants' choices manipulating both environmental richness and participants' needs, defined as the minimum number of points within a set of trials that a participant must earn to survive (Fig 1D). The results demonstrate that the optimal risk preference with respect to the survival probability changes depending on the specific environmental and need conditions. It reveals that, as the environment deteriorates and the need escalates, the optimal risk-aversiveness shifts towards risk-prone behavior (i.e., decreased risk-averseness towards $\alpha < 0$; Fig 1D, top). In contrast, with respect to the mean accumulated reward, being risk-neutral (i.e., risk-averseness $\alpha = 0$) is always optimal across varying environmental and need conditions (Fig 1D, bottom).

To test the foregoing predictions about the effect of environmental richness on participants' risk preference, we analyzed the participants' gambling tendencies on trials common to all environments ("Analyzed trials" in Fig 1B; see Methods for details).

First, we plotted the proportion of gambles accepted as a function of the reward magnitude and fitted sigmoid curves separately for the three reward probabilities (Fig 2A). We observed trends indicating that the participants accepted more gambles for the same set of choices when they were experienced in the poor than in the rich environment (Fig 2A, difference of the

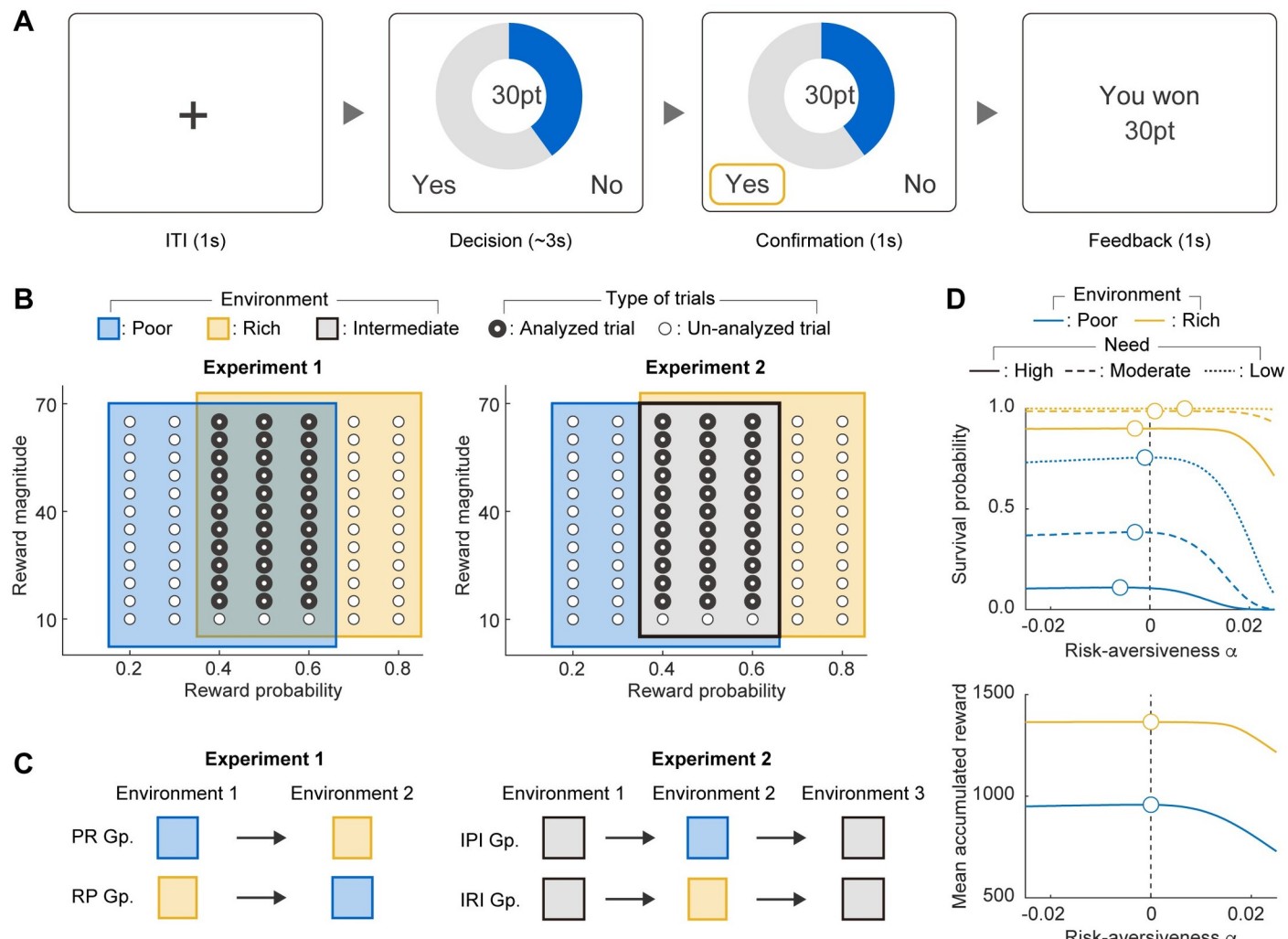

**Fig 1. Behavioral task and its settings. (A)** Timeline of a trial. On each trial, participants choose whether to accept or reject a risky gamble during the decision phase. The gamble magnitude (reward magnitude) was shown in the center and the probability of success (reward probability) was shown as the proportion of a pie chart shaded in blue. If the gamble was accepted, it led to a gain of the reward magnitude when successful and no gain if unsuccessful. Rejecting the gamble led to a sure gain of 10 points. Participant's choice was confirmed during the confirmation phase, and the points gained are shown during the feedback phase. **(B)** Settings of reward magnitudes and reward probabilities for Experiment 1 (left) and Experiment 2 (right). During the poor, rich, and intermediate blocks, participants were presented with gambles with parameters shown in the blue, yellow, and gray regions respectively. Note that participants were not informed about the presence of different environments. The order of trials was randomized for each participant. Only those trials with gamble parameters shown as the thick unfilled circles were used for the data analysis, and these gamble parameters were experienced in all the environments. Gambles with a reward magnitude of 10 were excluded from the analysis as they were catch trials (see Method for details). **(C)** Order in which environments were experienced in Experiment 1 (left) and Experiment 2 (right). In Experiment 1, one group experienced the poor environment followed by the rich environment (PR group), and the other group experienced the same environments in the reverse order (RP group). In Experiment 2, one group experienced the poor environment, and the other group experienced the rich environment in-between two intermediate environments (IPI group and IRI group, respectively). **(D)** Simulation of participants' choices. Participants' choices were generated using the risk-return model (Method), and risk-aversiveness $\alpha$ is plotted against survival probability (top) and mean accumulated reward (bottom) for the poor (blue) and rich (orange) environments and varying need (represented by different lines). In high, moderate, and low need conditions, the minimum number of points required for survival after 60 trials were set at 850, 1000, and 1150 points, respectively. The optimal risk preference for survival probability, indicated by the unfilled circle, varies with specific environmental and need conditions. As the environment worsens and needs increase, optimal risk-aversiveness shifts towards risk-prone behavior (decreases toward $\alpha<0$). In contrast, the optimal risk preference for mean accumulated reward remains constant across different environmental and need settings at risk-neutrality ($\alpha = 0$).

colors between the same line type), in accordance with our prediction. In both the first block of Experiment 1 (Fig 2A, left) and the second block of Experiment 2 (Fig 2A, right), the groups exposed to the poor environment accepted more gambles than those exposed to the rich

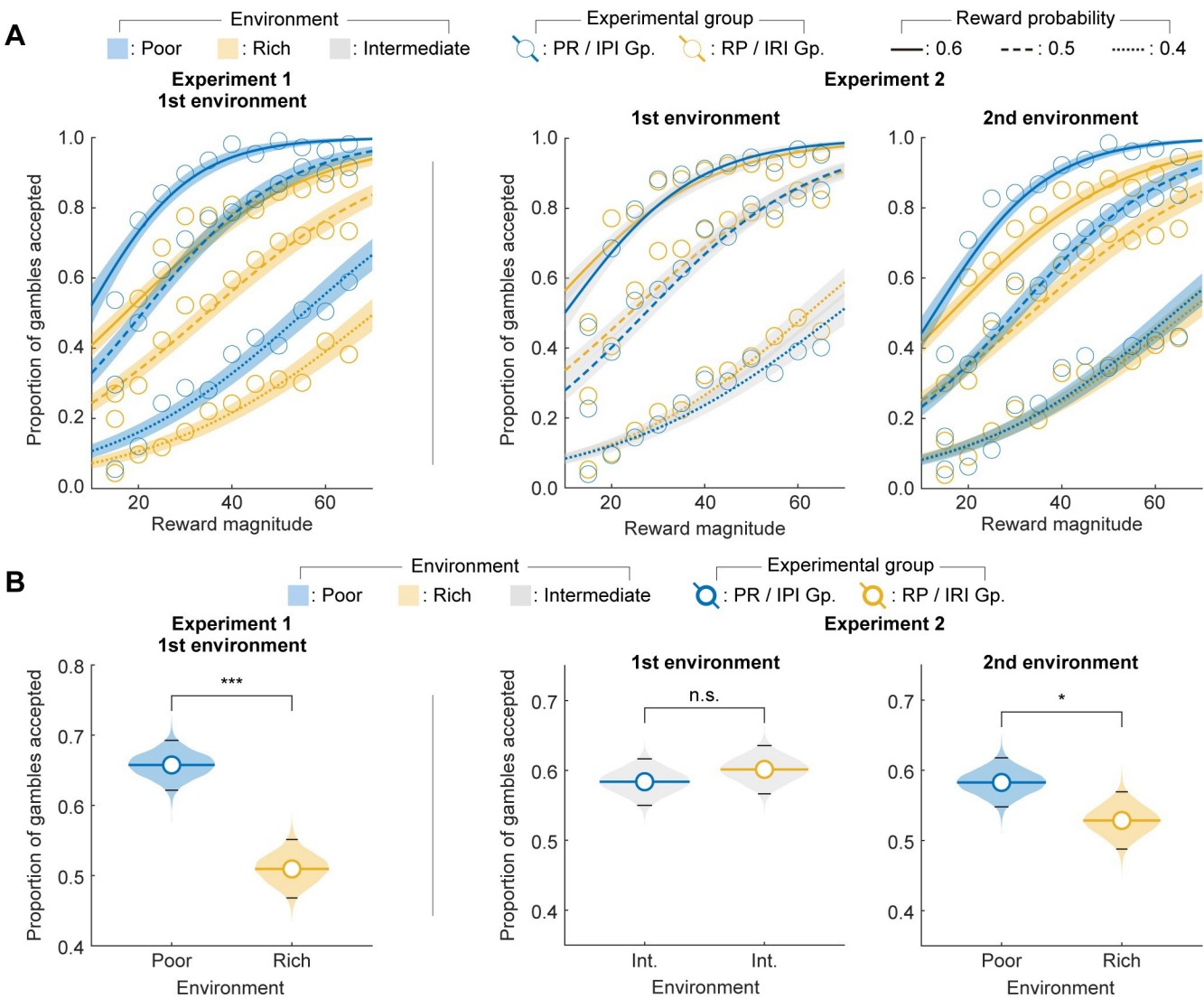

**Fig 2. Participants accepted significantly more gambles in the poor environment than in the rich environment.** (**A**) Proportion of gambles accepted for each combination of the gambling parameters. The color of the lines and circles represents the group, and the color of the shades represents the environment. Circles correspond to the proportion of gambles accepted by each group for each gamble presented. Lines are sigmoid curve fits to the proportion of gambles accepted by each group for the three reward probabilities experienced in all the environments. Shaded regions are one-standard deviation confidence intervals of the sigmoid curves computed by bootstrap. Separation of the orange and blue shaded lines for each reward probability shows that participants in the poor environment accepted more gambles than those in the rich environment. (**B**) Significance tests for the effect of environmental richness on gambling propensity. Choices shown in **A** were pooled for each environment and the proportion of gambles accepted were compared between environments (bootstrap test, one-tailed). The shaded regions represent the variation, due to random samplings of participants, in the mean of the proportion of gambles accepted as computed by bootstrap. The colored and black horizontal lines correspond to mean and 95% confidence intervals of the proportion of gambles accepted, respectively. $^{*}p < 0.05$; $^{***}p < 0.001$.

environment. In Experiment 2, the two groups did not seem to differ in the proportion of gambles accepted (Fig 2A, middle) during the first block, in which both groups experienced the intermediate environment.

Formal statistical tests showed that the differences observed in the fitted sigmoid curves in Fig 2A were significant. In the first block of Experiment 1, the PR group accepted significantly more gambles than the RP group ($p < 0.001$; unpaired bootstrap test, one-tailed; see Methods

for details; Fig 2B, left). Similarly, in the second block of Experiment 2, the IPI group accepted significantly more gambles than the IRI group ($p = 0.026$; unpaired bootstrap test, one-tailed; Fig 2B, right). In Experiment 2, to account for any differences in the initial risk tendencies and to quantify the effect of experiencing different environments on participants' risk preferences, we revisited the foregoing comparison by subtracting the proportion of accepted gambles in the initial block where both groups encountered the intermediate environment. We verified that the IPI group exhibited a significantly greater increase in the proportion of gambles accepted in the poor environment compared to the IRI group in the rich environment (S2 Fig; $p < 0.001$; unpaired bootstrap test, one-tailed). There was no significant difference between the two groups in the first block of Experiment 2 ($p = 0.762$; unpaired bootstrap test, one-tailed) (Fig 2B, middle). Qualitatively the same results were obtained from the participant-level analysis (S3 Fig).

## Previously experienced environments affect participants' risk preference

The foregoing findings show that the richness of the local environment in which an option is experienced affects participants' risk preference. In addition, the attractiveness of typical choice options in the current environment can be affected by whether environments experienced in the past were worse or better than the current environment. We hypothesize that worse past environments will increase, and better past environments will decrease the attractiveness of typical choice options in the current environment. Thus, we predict that prior exposure to either the rich or the poor environment will make participants more or less risk-averse, respectively. In the simulations of survival probability, the predicted shift in participants' risk preference is observed as a shift in the optimal risk preference towards risk-proneness (i.e., decreased risk-averseness towards $\alpha < 0$) as need increases (Fig 1D, top). Note that the accumulated reward after experiencing the rich environment is expected to be greater than that after experiencing the poor environment. Thus, to fulfill their needs, participants will be left with less to get (less need) after experiencing the rich as compared to the poor environment.

In Experiment 1, if the environment experienced in the first block does not affect participants' risk preference in the second block, then there should be no difference in the proportion of the gambles accepted by the two groups across the two blocks. In contrast, our hypothesis makes the prediction that the PR group should accept more gambles than the RP group across the two blocks. In support of our hypothesis, we found that the PR group accepted significantly more gambles than the RP group across the two blocks (Fig 3A, left; $p = 0.002$; unpaired bootstrap test, one-tailed). Furthermore, the foregoing difference between the PR and RP groups was significant for both the poor (Fig 3A, middle; $p < 0.001$, unpaired bootstrap test, one-tailed) and rich environment (Fig 3A, right; $p = 0.026$, unpaired bootstrap test, one-tailed).

In Experiment 2, the IPI and IRI groups experienced the poor and rich environment before the final block of intermediate environment, respectively. If the nature of the previously experienced environment has no effect on the risk preference in the subsequent environment, then there should be no difference between the choice tendencies of the two groups during the final block. In contrast, our hypothesis predicts that, while there will be downregulation in the proportion of gambles accepted during the final block as compared to the first block in both the groups (as the need will be lower than the initial need in both the groups), this downregulation will be small in the IPI group but much greater in the IRI group (because experiencing the rich environment in the second block will lower need more than experiencing the poor environment in the second block). While we found that the IPI

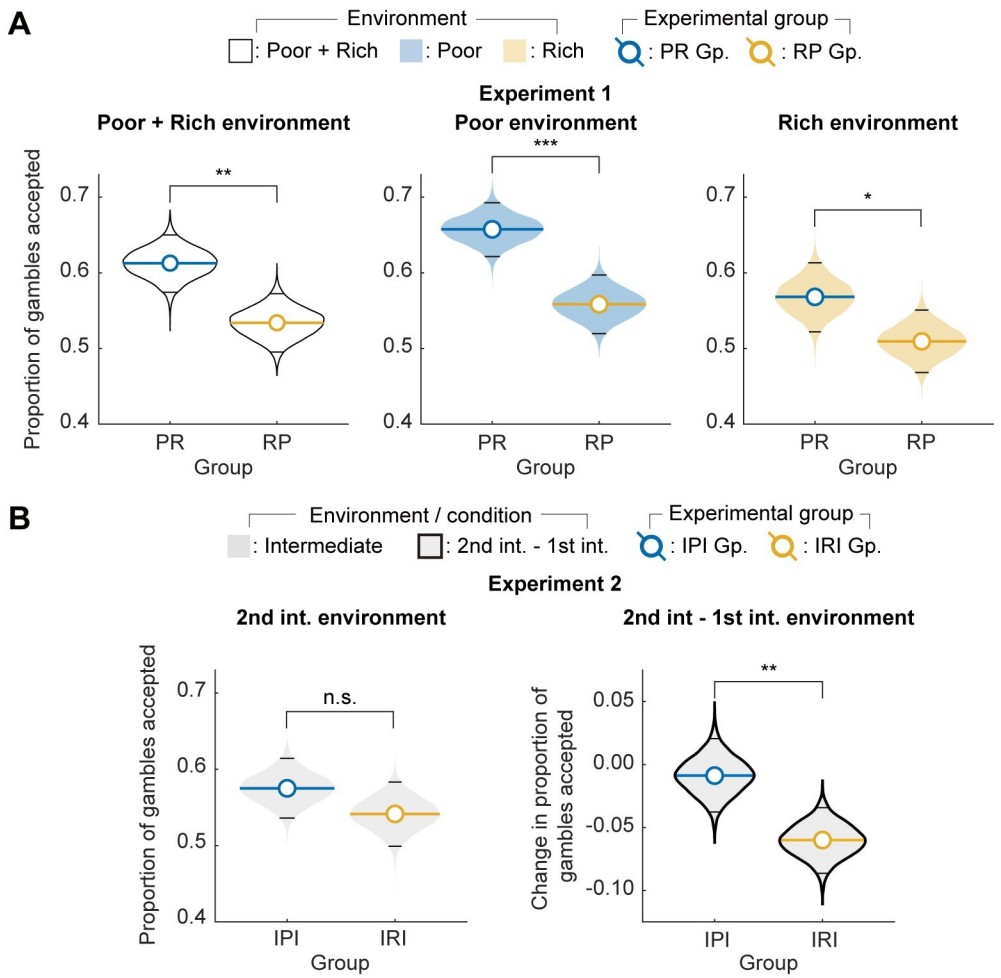

**Fig 3. Effects of the past environment on participants' risk preference.** (**A**) In Experiment 1, significantly more gambles were accepted when participants experienced the poor environment before the rich environment. The PR group accepted significantly more gambles than the RP group (left), in both the poor (middle) and rich (right) environments (bootstrap test, one-tailed). (**B**) In Experiment 2, while the IPI group accepted more gambles in the second intermediate environment than the IRI group, the direct comparison of the second intermediate environment did not reach statistical significance (left; bootstrap test, one-tailed). However, when comparing the change in the proportion of accepted gambles from the first to the second intermediate environment, the IRI group exhibited significantly larger downregulation in the proportion of gambles accepted than the IPI group (right; bootstrap test, one-tailed). The shaded regions represent the variation, due to random samplings of participants, in the mean of the proportion of gambles accepted as computed by bootstrap. The colored and black horizontal lines correspond to mean and 95% confidence intervals of the proportion of gambles accepted, respectively. $^*p < 0.05$; $^{**}p < 0.01$; $^{***}p < 0.001$.

group accepted more gambles than the IRI group during the final block, this difference did not reach significance (Fig 3B, right; $p = 0.125$, unpaired bootstrap test, one-tailed). However, as predicted, the downregulation in the proportion of gambles in the final block as compared to the first block was significantly greater in the IRI group compared to the IPI group (Fig 3B, left; $p = 0.006$; unpaired bootstrap test, one-tailed). We also observed qualitatively the same results when the same analysis was conducted at the participant level (S4 Fig).

## Fitting of computational models confirms the presence of the current and past environment effects

So far, our group-level model-free analysis have demonstrated that participants' risk preferences are influenced by both the richness of the current and past environment. Moving forward, we will delve into participant-level model-based analysis to examine whether the impact of the current and past environment can be elucidated as variations in the estimated values of model parameters.

To this end we employed a risk-return model and prospect theory-based models (see Methods for details). In brief, the risk-return model captures the risk preference of each participant with risk-aversiveness $\alpha$. We predicted $\alpha$ to be smaller during and after experiencing the poor environment as compared to the rich environment.

For the prospect theory-based models, we considered inclusion and exclusion of parameters in the value function $v(m_i)$ and the probability weighting function $\pi(p_i)$, where $m_i$ and $p_i$ represent the reward magnitude and reward probability in the $i$-th trial, respectively. The parameters $\lambda$ and $\gamma$ in the value function and probability weighting function, respectively, are both risk-proneness parameters (see Methods for details). Hence, we predicted these parameters to be greater during and after experiencing the poor environment compared to the rich environment. Additionally, we considered utility functions in two forms: $v(m_i)\pi(p_i)$ (multiplicative model) and $(1 - w_p)v(m_i) + w_p\pi(p_i)$ (additive model). The parameter $w_p$ in the additive model serves as the weighting parameter for reward probability, interpretable in our experimental context as a risk-aversiveness parameter, promoting the participants to reject more gambles. Therefore, we predicted $w_p$ to be smaller during and after experiencing a poor environment compared to a rich environment. In summary, our analysis encompassed the risk-return model, along with four variants (involving considerations of both inclusion and exclusion of the parameters $\lambda$ and $\gamma$) of the multiplicative model and the additive model. A sigmoid choice rule was commonly employed for all the utility models. All computational models were then fitted to the trial-by-trial choices of each participant for each environment using a Bayesian hierarchical expectation-maximization method (Methods).

The model comparison with integrated Bayesian information criterion (iBIC) showed that the best model was the additive model including both $\lambda$ and $\gamma$ (S1 Table). Within the multiplicative model, the best model was also the model including both $\lambda$ and $\gamma$. Among the models with two parameters, the risk-return model (that includes risk-aversiveness $\alpha$ and choice stochasticity $\beta$) was the best model. Overall, the mean estimated value of $\lambda$ in the additive and multiplicative models were below one, and the mean estimated value of $\alpha$ in the risk-return model was above one for all environments across two experiments, indicating participants' inclination to risk-averse choice tendencies (S5 Fig).

We tested our predictions (one-tailed t-tests in the direction of our prediction with Bonferroni correction) regarding the changes in risk-preference for each model parameter of the best performing models separately (i.e., $\alpha$ for the risk-return model with, $\lambda$ and $\gamma$ for the multiplicative model, $\lambda$, $\gamma$ and $w_p$ for the additive model; S2 Table). The predicted changes in risk-preference parameters were significant for the risk-aversiveness $\alpha$ of the risk-return model, consistently across Experiment 1 and Experiment 2. This result underscores the influence of both the current and past environmental effects on participants' risk preferences.

Additionally, we also fitted the foregoing models by replacing the sigmoid choice rule with an approach-avoidance choice rule (see Methods). In these models, the risk-proneness parameter $\eta$ biases the probability of gambling by changing the maximum (if $\eta<0$) or minimum (if $\eta>0$) probabilities to gamble in the sigmoid curve. The parameter $\eta$ serves as a risk-proneness

parameter, and, consequently, we predicted it to be greater during and after experiencing the poor environment compared to the rich environment.

The model comparison of the approach-avoidance version of the models with iBIC showed that the best model was the additive model including both $\lambda$ and $\gamma$ (S1 Table). Within the multiplicative models, the best model was also the one including both $\lambda$ and $\gamma$. Among the models with three parameters, the risk-return model (that includes $\alpha$, $\beta$, and $\eta$) was the best model. Lastly, model comparison between the approach-avoidance choice-based models and sigmoid choice-based models were inconclusive, such that with the extra model parameter $\eta$, iBIC improved for the risk-return model, but not for the multiplicative and additive models.

We tested our predictions (one-tailed t-tests in the direction of our prediction with Bonferroni correction) regarding the changes in risk-preference for each model parameter of the best performing models separately (i.e., $\alpha$ and $\eta$ for the risk-return model, $\lambda$, $\gamma$ and $\eta$ for the multiplicative model, $\lambda$,$\gamma$,$w_P$ and $\eta$ for the additive model; S3 Table). The predicted changes in risk-preference parameters were significant for the risk-proneness $\eta$ of the risk-return model, and the risk-proneness $\gamma$ of the multiplicative model, and the risk-proneness $\eta$ of the additive model, consistently across Experiment 1 and Experiment 2. This result confirms again the influence of both the current and past environmental effects on participants' risk preferences.

## Temporally resolved analysis confirm the current and past environment effects

To gain more insight into the dynamics by which participants' risk preference are modulated by the current and past environments, we sought to capture the trial-by-trial fluctuations in the proportion of gambles accepted within each group across two experiments. To this end, we calculated a moving average of the proportion of gambles accepted for each group in both experiments (Fig 4A). We considered the preceding and succeeding two trials for any given trial, resulting in a window size of 5 trials, for computing the moving average of the proportion of gambles accepted (Methods).

The observed time-course of the proportion gamble accepted across the experiments confirmed the presence of the current environment effect (Fig 4B, top) as well as the past environment effect (Fig 4B, bottom). With respect to the current environment, in the first block of Experiment 1, the mean proportion of gambles accepted was significantly higher for the PR group ($M = 0.66$, $SD = 0.03$), which encountered the poor environment, than the RP group ($M = 0.51$, $SD = 0.04$; unpaired t-test, one-tailed; $t(130) = 23.6$, $p < 0.001$), which encountered the rich environment. Similarly, in the second block of Experiment 2, the mean proportion of gambles accepted was significantly higher for the IPI group ($M = 0.58$, $SD = 0.03$), which encountered the poor environment, than the IRI group ($M = 0.53$, $SD = 0.03$; unpaired t-test, one-tailed; $t(130) = 10.0$, $p < 0.001$), which encountered the rich environment. These results are consistent with the effect of the current environment seen in the block-wise analysis of the proportion of gambles accepted (Fig 2B).

With respect to the past environment, across Experiment 1, the mean proportion of gambles accepted by the PR group ($M = 0.61$, $SD = 0.06$) was higher than that accepted by the RP group ($M = 0.53$, $SD = 0.04$; unpaired t-test, one-tailed; $t(262) = 12.9$, $p < 0.001$). In addition, in both the poor and rich environments, the mean proportion of the gamble accepted by the PR group (poor: $M = 0.66$, $SD = 0.03$, rich: $M = 0.57$, $SD = 0.04$) was higher than that by the RP group (poor: $M = 0.56$, $SD = 0.03$, rich: $M = 0.51$, $SD = 0.04$; unpaired t-test, one-tailed; poor: $t(130) = 21.5$, $p < 0.001$, rich: $t(130) = 8.2$, $p < 0.001$). For Experiment 2, the mean proportion of gambles accepted by the IPI group during the final block ($M = 0.58$, $SD = 0.02$) was significantly higher than that accepted by the IRI group ($M = 0.54$, $SD = 0.02$; unpaired t-test, one-

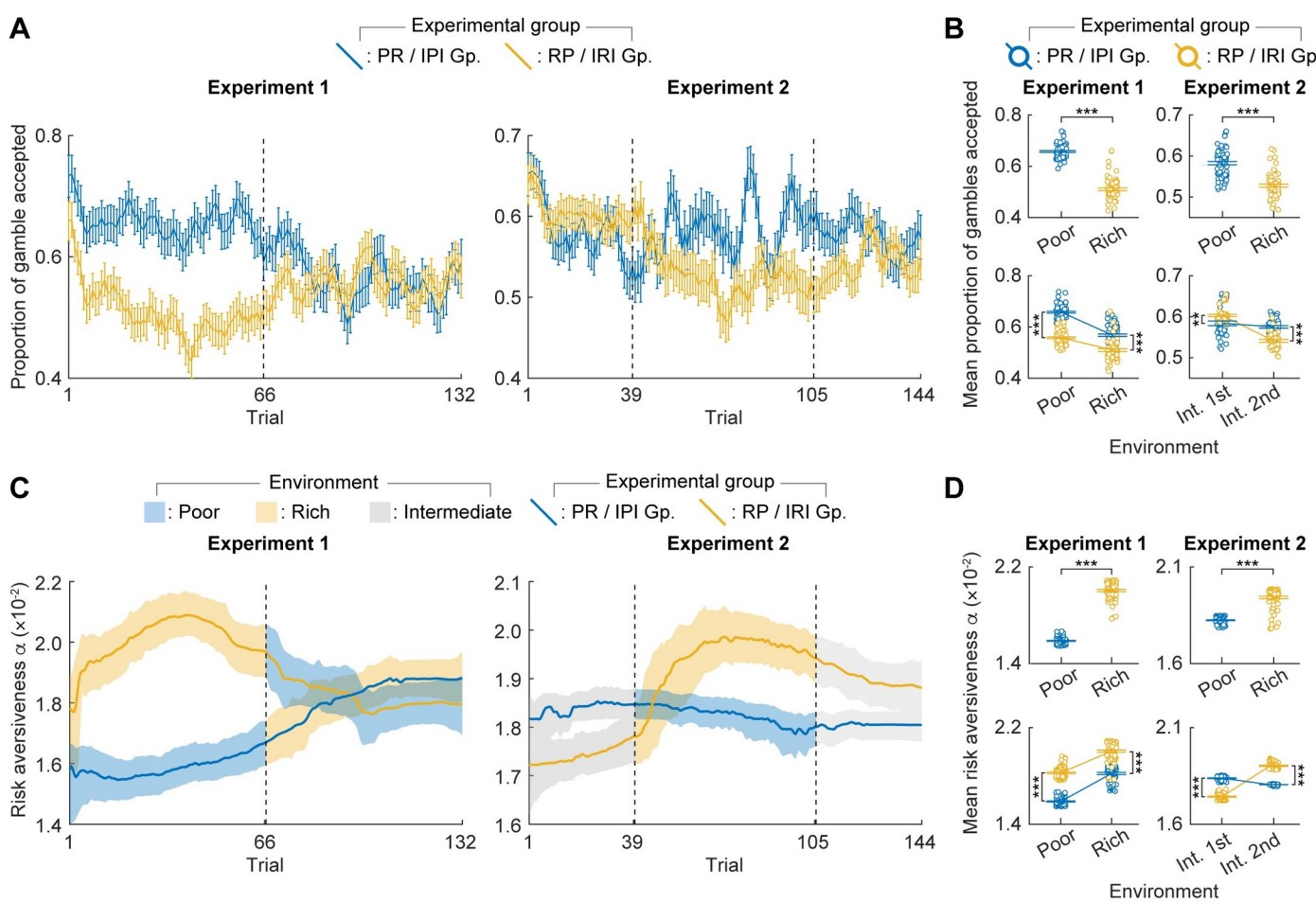

**Fig 4. Dynamics in participants' risk preference across the experiment.** (**A**) A moving average of the proportion of gambles accepted is computed for each group across Experiment 1 (left) and Experiment 2 (right). Each data point represents the proportion of gambles accepted by participants in a given trial, along with the two preceding and two succeeding trials, forming a window size of 5 trials. Vertical bars represent standard error. (**B**) The mean proportions of gambles accepted in each environment by each group. In both experiments, the mean proportion of gambles accepted were significantly higher when participants are experiencing the rich as compared to the poor environment ($p < 0.001$ for Experiment 1 and 2; top). In addition, the mean proportions of gambles accepted were significantly higher after participants experienced the poor than the rich environment ($p < 0.001$ for poor & rich environment in Experiment 1 and for the second intermediate environment in Experiment 2; bottom). One-tailed unpaired t-tests with Bonferroni correction were used for statistical comparisons of the proportion of gambles accepted (except for the comparison of the first intermediate in Experiment 2 where two-tailed test was employed given the initial prediction that there is no difference.) Horizontal bars and unfilled dots represent the mean and raw values of the proportion of gambles accepted, respectively. (**C**) Risk-aversiveness $\alpha$ of each group estimated by a particle filter. Shaded regions are 90% confidence intervals for risk-aversiveness $\alpha$. (D) Mean risk-aversiveness $\alpha$ estimated by the particle filter is plotted mirroring the format of (B). In both experiments, participants were significantly more risk-averse when experiencing the rich as compared to the poor environment ($p < 0.001$ for Experiment 1 and 2; top). In addition, participants were significantly more risk-averse after experiencing the rich rather than the poor environment ($p < 0.001$ for the poor & rich environment in Experiment 1 and for the second intermediate environment in Experiment 2; bottom). **$p < 0.01$; ***$p < 0.001$.

tailed; $t(76) = 7.3$, $p < 0.001$). These results are consistent with the effect of the past environment seen in the block-wise analysis of the proportion of gambles accepted (Fig 3). Additionally, the mean proportion of the gamble accepted in the first intermediate environment of Experiment 2 was significantly different between the IPI group ($M = 0.58$, $SD = 0.04$) and the IRI group ($M = 0.60$, $SD = 0.02$; unpaired t-test, two-tailed; $t(76) = -3.0$, $p = 0.003$). Considering that both groups experienced the same environment at the start of the experiment, the observed difference is likely attributable to variations in participant sampling. This interpretation aligns with our earlier observation, wherein direct comparisons of the proportion of

gambles accepted in the second block (Fig 2B, right) and the third block (Fig 3B, left) showed improvement when the proportion of accepted gambles in the first block is subtracted (baseline subtraction: Figs S2 and 3B, right).

Next, we estimated the risk preference of each group using a state-space model. In brief, we assumed that each group made risky choices based on a temporally fluctuating and correlated risk-aversiveness and estimated its time-course from participants' choices using a particle filter (Fig 4C). In contrast to the assumption of independence made in the analysis of the proportion of gambles accepted, this method operates under the presumption of temporal correlation in participants' risk-aversiveness. It estimates the participants' risk preference not solely from choices at the current time point but also leverages information from temporally proximal trials. The overall flexibility of the dynamics is optimized by examining the choice data across the entire experiment (see Methods for details). The risk-aversiveness estimated by the particle filter showed significant negative correlation with the proportion of the gamble accepted (Experiment 1: $r = -0.91$, $p < 0.001$; Experiment 2: $r = -0.77$, $p < 0.001$).

The risk-aversiveness estimated by the particle filter yielded quantitatively consistent results with the time-resolved analysis of the proportion of gambles accepted, highlighting the effects of the current environment (Fig 4D, top) and past environments (Fig 4D, bottom) on participants' risk preference. In the first block of Experiment 1 and in the second block of Experiment 2, participants were significantly more risk-averse when experiencing the rich (Experiment 1: $M = 2.00\times10^{-2}$, $SD = 0.07\times10^{-2}$, Experiment 2: $M = 1.94\times10^{-2}$, $SD = 0.05\times10^{-2}$) as compared to the poor environment (Experiment 1: $M = 1.59\times10^{-2}$, $SD = 0.03\times10^{-2}$; unpaired $t$-test, one-tailed; $t(130) = 41.6$, $p < 0.001$; Experiment 2: $M = 1.82\times10^{-2}$, $SD = 0.02\times10^{-2}$; unpaired $t$-test, one-tailed; $t(130) = 16.5$, $p < 0.001$). This shows that participants were consistently more risk-averse in the rich than in the poor environment across both the experiments, highlighting the effect of the current environment. Furthermore, in Experiment 1, the mean risk-aversiveness estimated by the particle filter was consistently higher for the RP group than for the PR group across the entire experiment (RP: $M = 1.91\times10^{-2}$, $SD = 0.11\times10^{-2}$, PR: $M = 1.70\times10^{-2}$, $SD = 0.13\times10^{-2}$; unpaired $t$-test, one-tailed; $t(262) = 14.4$, $p < 0.001$), in the poor environment (RP: $M = 1.83\times10^{-2}$, $SD = 0.05\times10^{-2}$, PR: $M = 1.59\times10^{-2}$, $SD = 0.03\times10^{-2}$; unpaired $t$-test, one-tailed; $t(130) = 33.4$, $p < 0.001$), and in the rich environment (RP: $M = 2.00\times10^{-2}$, $SD = 0.07\times10^{-2}$, PR: $M = 1.82\times10^{-2}$, SD $= 0.06\times10^{-2}$; unpaired $t$-test, one-tailed; $t(130) = 15.0$, $p < 0.001$). In Experiment 2, the mean risk-aversiveness during the final block was significantly higher for the IRI group ($M = 1.90\times10^{-2}$, $SD = 0.02\times10^{-2}$) than for the IPI group ($M = 1.80\times10^{-2}$, $SD = 0.003\times10^{-2}$; unpaired $t$-test, one-tailed; $t(76) = 32.5$, $p < 0.001$). These observations again highlight the effect of the past environment on participants' risk preference.

The performance of the particle filter estimation was verified by model recovery analysis (S6 Fig). To further validate the estimated risk-aversiveness by the particle filter, we simulated choice data based on the estimated values of risk-aversiveness and computed the proportion of the gambles accepted across the experiments (S7 Fig A). The proportions of gambles accepted for the simulated choices showed significant positive correlation with that of participants' choices (Experiment 1: $r = 0.63$, $p < 0.001$; Experiment 2: $r = 0.52$, $p < 0.001$), and we observed quantitively consistent results with respect to the effects of the current and past environments (S7 Fig B).

Our findings from both the proportion of gambles accepted and the risk-aversiveness estimated by the particle filter support our hypothesis that the nature of the current and past environment affects participants' risk preference.

### Environment-related variables affect participants' risk preference

The results so far show that participants' risk preference depends on both the current environment and the past environments. We next performed a regression analysis to elucidate which experimental variables affected participants' risk preference.

First, we predicted that the attractiveness of the same probability option will be higher in the poor environment than in the rich environment, and this difference in attractiveness will make the participants more risk-prone in the poor environment. We coded local attractiveness regressor as the deviation of reward probability of the current option from the mean reward probability of all gambling options in the environment ($\Delta p_c$). Since the mean reward probability was lower in the poor environment than in the rich environment, $\Delta p_c$ for the choice option with the same reward probability was always higher in the poor environment than in the rich environment, and we expected this variable to be positively correlated with participants' gambling choices.

Second, we predicted that typical choice options in an environment seem less attractive when the environment is worse than the previously experienced environment and vice versa. We coded the regressor for the effect of past environments on the attractiveness of choice options in the current environment as the difference between the mean reward probability in the current and past environments ($\Delta p_p$). Thus, $\Delta p_p$ is negative when the current environment is worse than the past environment and vice versa. We expected $\Delta p_p$ to be positively correlated with participants' gambling choices.

Additionally, we presumed that the attractiveness of the same option diminishes when the total reward accumulated until the current trial is higher. To account for this, we introduced a regressor of the total reward accumulated until the current trial as another regressor (Acc. reward). This additional regressor, alongside the $\Delta p_c$ and $\Delta p_p$, may capture the influence of the current and past environmental effects. Specifically, it explores the possibility that participants exhibit greater risk aversion during and after experiencing the rich environment compared to the poor environment, as it is on average easier to accumulate more reward in the former environment. We expected Acc. reward to be negatively correlated with participants' gambling choices.

We entered $\Delta p_c$ and $\Delta p_p$ as regressors of interest to capture the effects of current and past environmental statistics, respectively. To control for the effect of the properties of the choice option on the gamble acceptance rate, we included expected value (EV) and Risk as control regressors. We expected participants to gamble more for higher EVs and gamble less for riskier options. In addition, to control for the effect of the previous trial, we included regressors for previous reward (Prev. reward), previous choice (Prev. choice) and their interaction (Prev. success).

We ran a logistic regression on participants' gambling choices with the foregoing eight regressors: EV, Risk, $\Delta p_c$, $\Delta p_p$, Prev. choices, Prev. reward, Prev. success, and Acc. reward (Fig 5; see Methods for details). This model showed the lowest value of BIC (Experiment 1: 16299, Experiment 2: 27756), outperforming the model only with EV and Risk (Experiment 1: 17094, Experiment 2: 29204), the model with EV, Risk, $\Delta p_c$, and $\Delta p_p$ (Experiment 1: 16663, Experiment 2: 28843), and the model with EV, Risk, Prev. choice, Prev. reward, Prev. success, and Acc. reward (Experiment 1: 16819, Experiment 2: 28217).

For the full model, as expected, we observed significantly positive and negative regression coefficients for EV and Risk consistently for all groups across the two experiments (Experiment 1: $\beta_{EV} = 1.938, p = 0.002, \beta_{Risk} = -0.876, p = 0.002$; Experiment 2: $\beta_{EV} = 2.135, p = 0.002, \beta_{Risk} = -1.056, p = 0.002$; one-sample bootstrap test against zero, two-tailed with Bonferroni correction).

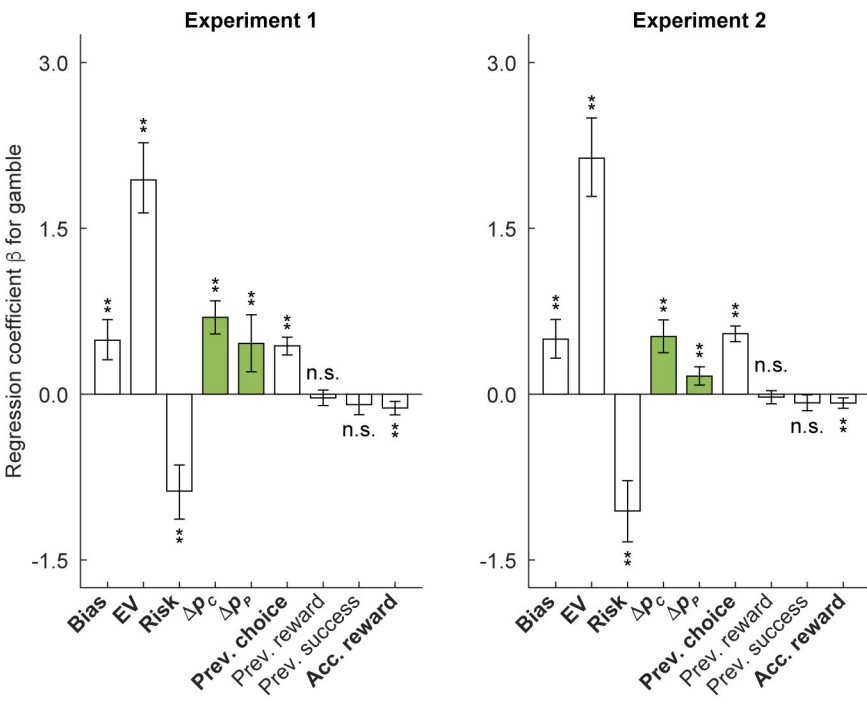

**Fig 5. Regressors capturing the relative attractiveness of a gambling option compared to other options in the current and past environments play a significant role in explaining participants' gambling propensity.** Participants' gambling propensity was positively correlated with the deviation of reward probability of the current gambling option from the mean reward probability of all gambling options in the current environment ($\Delta p_c$). Participants' gambling propensity was also positively correlated with the difference between the mean reward probability in the current and past environments ($\Delta p_p$). Additionally, participants' gambling propensity was negatively correlated with the accumulated rearward (Acc. reward). Regressors with consistently significant regression coefficients across the two experiments are noted in bold. Error bars represent 95% confidence bounds computed from bootstrap. Two-tailed one-sample bootstrap tests against zero with Bonferroni correction were used for statistical tests of regression coefficients. $^*p < 0.05$; $^{**}p < 0.01$.

In particular, as predicted, the regression coefficient for $\Delta p_c$ was significantly positive across the two experiments (Fig 5; Experiment 1: $\beta_{\Delta p_c} = 0.695, p = 0.002$; Experiment 2: $\beta_{\Delta p_c} = 0.521, p = 0.002$), suggesting that $\Delta p_c$ made a significant contribution to the participants' tendency to gamble more when the same choice option was experienced in the poor than in the rich environment. The regression coefficient for $\Delta p_p$ was also significantly positive across the two experiments (Fig 5; Experiment 1: $\beta_{\Delta p_p} = 0.459, p = 0.007$; Experiment 2: $\beta_{\Delta p_p} = 0.163, p = 0.002$), suggesting that $\Delta p_p$ made a significant contribution to the participants' tendency to gamble more when the environment in which the choice was being made was better than the previously experienced environment and vice versa.

In addition, the regression coefficients for Prev. choice were significantly positive across the two experiments (Fig 5; Experiment 1: $\beta_{\text{Prev.choice}} = 0.438, p = 0.002$; Experiment 2: $\beta_{\text{Prev.choice}} = 0.547, p = 0.002$). The regression coefficients for Prev. reward and Prev. success were not significant across the two experiments (Fig 5; Experiment 1: $\beta_{\text{Prev.reward}} = -0.033, p = 1.000, \beta_{\text{Prev.success}} = -0.094, p = 0.403$; Experiment 2: $\beta_{\text{Prev.reward}} = -0.027, p = 1.000, \beta_{\text{Prev.success}} = -0.079, p = 0.284$). Lasty, the regression coefficient for Acc. reward was significantly negative for Experiment 1 and Experiment 2 (Fig 5;

Experiment 1: $\beta_{\text{Acc.Reward}} = -0.125, p = 0.002$; Experiment 2: $\beta_{\text{Acc.reward}} = -0.08, p = 0.007$), suggesting that Acc. reward made a significant contribution in that participants' tended to gamble less when the accumulated reward is high.

Several regressors in our analysis were correlated and had a high variance inflation factor (VIF; S4 Table). Although the bootstrapped confidence intervals of the regression coefficients indicated a reasonable stability in their estimated values, we conducted several additional tests to investigate the potential influence of multicollinearity. Specifically, we reran the regression with L1 regularization, which robustly estimates regression coefficients in the presence of the multicollinearity. In addition, we reran the regression with i) only the primary regressors (i.e., the regressors other than EV, Risk, $\Delta p_c$, and $\Delta p_p$ are excluded), and ii) excluding regressors that exhibited a VIF greater than five in either experiment (i.e., the regressors for EV, Risk, and Prev. success were excluded; S8 Fig). The estimated coefficients from both the L1 regularized regression and the regression with only the primary regressors showed no significant differences compared to those estimated from the original regression (one-sample bootstrap test against the estimated value in the original model, two-tailed with Bonferroni correction, all $p > 0.05$). Significant differences were observed only in the regression that excluded regressors with high VIFs. Specifically, the regression coefficient for $\Delta p_c$ was significantly different for both Experiment 1 and Experiment 2 (both $p = 0.001$), and that for $\Delta p_p$ was significantly different in Experiment 2 ($p = 0.001$). The elevated values of these regression coefficients are likely due to the removal of the EV regressor, which exerted the largest influence on participants' choices and showed positive association with $\Delta p_c$ and $\Delta p_p$. Nevertheless, the estimated values of the coefficients consistently exhibited the same signs in all regression analyses, and the overall results confirm a reasonable stability of the regression coefficients estimated from our original model.

In summary, the regression results support our findings from the analysis of the proportion of gambles accepted and from the estimation of participants' risk aversion parameter using particle filter (see previous sections), that both the richness of the current environment as well as the relative nature of the past environments affect participants' risk preferences.

## Environment-related variables are necessary to explain participants' risk preference dynamics

Next, we simulated participants' choices with significant regressors common to both experiments (shown in bold in Fig 5), and computed the proportion of gambles accepted (simulation 1; see Methods), as we did for the real data (Fig 4A and 4B). To verify the importance of environment-related variables $\Delta p_c$ and $\Delta p_p$ in explaining the participants' risk preference dynamics, we ran another simulation excluding the two environment-related regressors (simulation 2). For this simulation, the regression model without the two environment-related variables was fitted to the participants' choices for each experiment. The resulting estimated regression coefficients were then employed for the simulation of choice data (S9 Fig).

In simulation 1 (Fig 6A and 6B), the proportion of gambles accepted was positively and significantly correlated with that observed in the real choices (Experiment 1: $r = 0.71, p < 0.001$; Experiment 2: $r = 0.33, p < 0.001$). Furthermore, we observed qualitative consistency between the proportion of gambles accepted computed for the simulated choices and the real choices for both Experiment 1 and Experiment 2, recovering the effects of the current environment (Fig 6B, top) and past environments (Fig 6B, bottom). In the first block of Experiment 1 and in the second block of Experiment 2, the mean proportions of gambles accepted in the simulated choices were significantly higher in the poor (Experiment 1: $M = 0.64, SD = 0.02$, Experiment 2: $M = 0.60, SD = 0.03$) as compared to the rich environment (Experiment 1: $M = 0.50$,

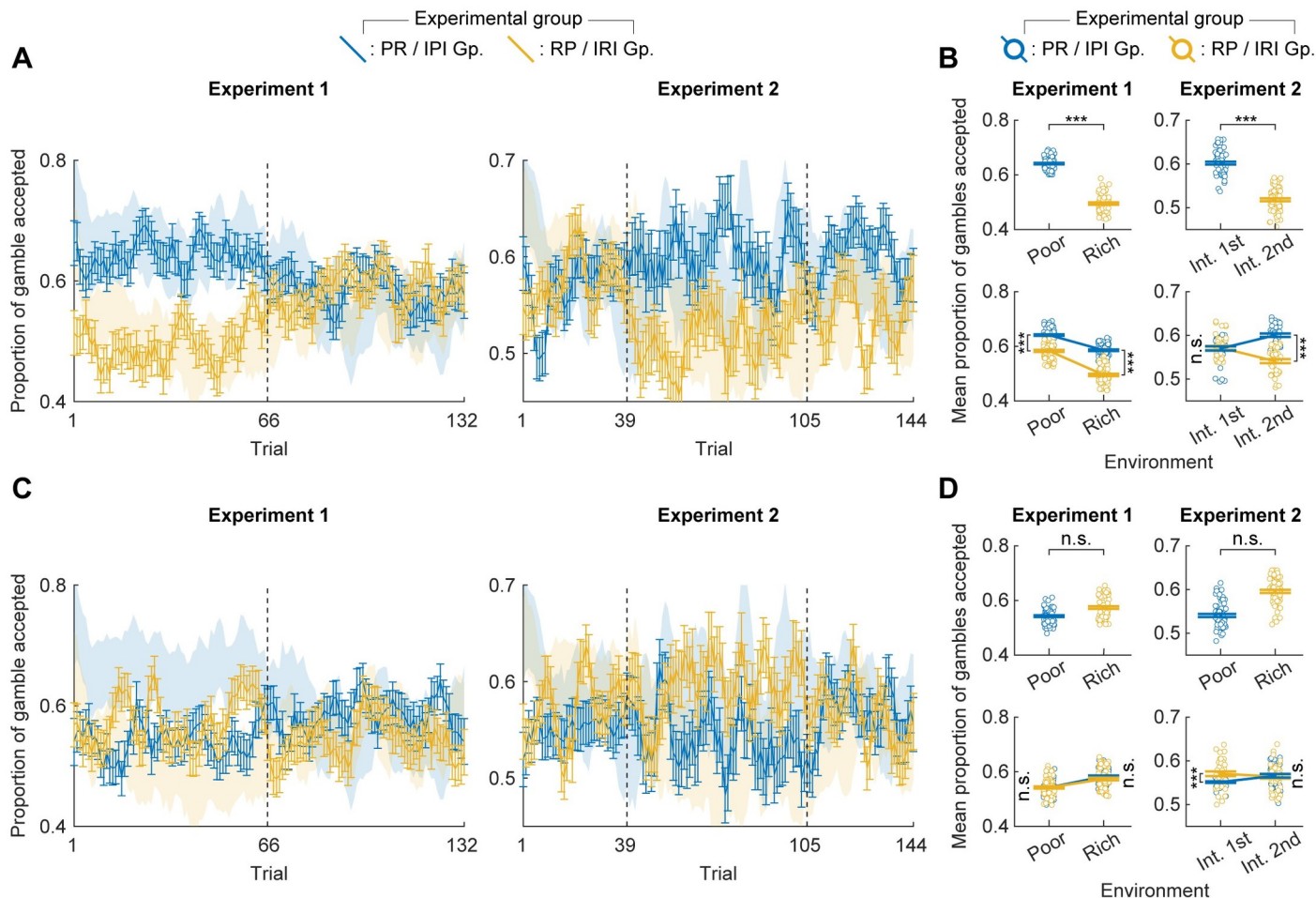

**Fig 6. Environment-related regressors are necessary in recovering the observed dynamics in risk preference.** (**A, B**) Regressors shown in bold in Fig 5 and their coefficients were used to simulate participants' gambling choices (simulation 1, $n = 120$ for each group). The moving average of the proportion of gambles accepted for each trial (A) and their environment-wise averages (B) were calculated for each group. The resulting dynamics showed close similarities with that computed from the real data (Fig 4A), recovering the effect of the current (B top) and past (B bottom) environments. (**C, D**) The dynamics of the proportion of gambles accepted present in the real data could not be recovered when the environment-related regressors $\Delta p_c$ and $\Delta p_p$ are excluded (simulation 2, the regressors shown in bold in S9 Fig and their coefficients were used to simulate participants' gambling choices; $n = 120$ for each group). (**A, C**) Error bars represent standard errors. Shaded regions are 95% confidence intervals of the proportion of gambles accepted computed for the real data. (**B, D**) Horizontal bars and unfilled dots represent the mean and raw values of the proportion of gambles accepted, respectively. One-tailed unpaired t-tests with Bonferroni correction were used for statistical comparisons of the mean proportion of gambles accepted (except for the comparison for the first intermediate in Experiment 2 where a two-tailed test was employed given the initial prediction that there is no difference.) ***$p < 0.001$.

$SD = 0.03$; unpaired $t$-test, one-tailed; $t(130) = 29.7$, $p < 0.001$; Experiment 2: $M = 0.52$, $SD = 0.03$; unpaired $t$-test, one-tailed; $t(130) = 18.1$, $p < 0.001$). Furthermore, in Experiment 1, the mean proportion of gambles accepted were consistently higher in the simulated choices of PR group than for the RP group across the entire experiment (PR: $M = 0.61$, $SD = 0.04$, RP: $M = 0.54$, $SD = 0.05$; unpaired $t$-test, one-tailed; $t(262) = 13.0$, $p < 0.001$), in the poor environment (PR: $M = 0.64$, $SD = 0.02$, RP: $M = 0.58$, $SD = 0.03$; unpaired $t$-test, one-tailed; $t(130) = 12.7$, $p < 0.001$), and in the rich environment (PR: $M = 0.59$, $SD = 0.02$, RP: $M = 0.50$, $SD = 0.03$; unpaired $t$-test, one-tailed; $t(130) = 17.6$, $p < 0.001$). In Experiment 2, the mean proportion of gambles accepted for the simulated choices during the final block were significantly higher for the IPI group ($M = 0.60$, $SD = 0.03$) than for the IRI group ($M = 0.54$,

$SD = 0.03$; unpaired $t$-test, one-tailed; $t(76) = 9.5$, $p < 0.001$). All the foregoing simulation results (simulation 1) are in accordance with the mean proportion of gambles accepted observed in the participants' real choices (Fig 4A and 4B).

In contrast, in simulation 2, where the environment-related regressors $\Delta p_c$ and $\Delta p_p$ were excluded (Fig 6C and 6D), the observed proportion of gambles accepted were negatively correlated with that observed in the real choices (Experiment 1: $r = -0.25$, $p < 0.001$; Experiment 2: $r = -0.35$, $p < 0.001$). Furthermore, we did not observe qualitative consistency between the mean proportion of gambles accepted computed for the simulated choices and the real choices both for Experiment 1 and Experiment 2, failing to recover the effects of the current environment (Fig 6D, top) and past environments (Fig 6D, bottom). In particular, the mean proportion of gambles accepted were higher during the rich than during the poor environment in the first block of Experiment 1 (poor: $M = 0.54$, $SD = 0.03$, rich: $M = 0.57$, $SD = 0.04$) and in the second block of Experiment 2 (poor: $M = 0.54$, $SD = 0.03$, rich: $M = 0.60$, $SD = 0.03$), contrary to our prediction and the trends observed in the real data (Fig 4A and 4B).

To further assess the necessity of the environment-related regressors in explaining the dynamics in participants' risk preferences, we generated simulated choices by systematically excluding each of the following regressors from the original model in separate iterations: $\Delta p_c$, $\Delta p_p$, Prev. choice, and Acc. reward (S10 Fig). In the model where $\Delta p_c$ was excluded (S10A and S10B Fig), the impact of the current environment deviated from the expected pattern (S10B Fig, top), while the influence of the past environment aligned with the observed dynamics in the real choices (S10B Fig, bottom). This observation supports the notion that $\Delta p_c$, which reflects the attractiveness of an option relative to others in the local environment, plays a critical role in explaining the effect related to the current environment.

Next, in the model where $\Delta p_p$ was excluded (S10C and S10D Fig), the effect of the current environment remained consistent with the observed dynamics in real choices (S10D Fig, top), while the impact of the past environment was notably attenuated (S10D Fig, bottom). This observation supports the notion that $\Delta p_p$, which reflects the attractiveness of an option relative to others in the past environment, plays a crucial role in explaining the effect related to the past environment.

Finally, contrasting with the preceding two models, in the models where Prev. choice (S10E Fig) or Acc. reward (S10G Fig) was excluded, the observed patterns concerning both the current environment (S10F Fig, top and S10H Fig, top) and past environment (S10F Fig, bottom and S10H Fig, bottom) remained in alignment with the dynamics observed in real choices.

Additionally, considering the relatively high values of the regression coefficients for the Prev. choice regressor (Fig 5), we conducted a subsequent regression analysis. In this iteration, the model incorporated not only the original set of regressors but also included additional regressors for choices made at two and three trials ago, and the moving average of the proportion of gambles accepted across the experiments are computed for the choice generated (S11A Fig). We confirmed qualitative consistency between the proportion of gambles accepted computed for the simulated choices and the real choices for both Experiment 1 and Experiment 2, again recovering the effects of the current environment (S11B Fig, top) and past environments (S11B Fig, top).

In summary, the simulation results show that the environment-related regressors ($\Delta p_c$ and $\Delta p_p$) are necessary for recovering the risk preference patterns observed in the participants' choices.

Thus, all our results, from proportion of gambles accepted, particle filter estimation and logistic regression for the real and simulated data, show that the current and past environments play important roles in shaping a person's risk preference. We suggest the relative

attractiveness of a gambling option compared to the other options in the local environment and compared to the past environments as a possible underlying psychological mechanism.

## Discussion

In this study, we investigated the effect of environmental richness on participants' risk preference across two experiments. Phenomenologically, our findings were two-fold. First, we found that the participants were consistently more risk-averse in the rich as compared to the poor environment for the same set of choice options. Thus, the richness of the current environment affected participants' risk preference. Second, participants were more risk-averse in an environment of intermediate richness after experiencing the rich as compared to the poor environment in the prior block. This shows that the richness of the past environment relative to the current environment influenced participants' risk preference–a deterioration in the environmental richness rendered participants' more risk-averse and vice versa. Regression analysis showed that two regressors capturing the effects of the current and past environmental richness respectively were necessary to explain the participants' risky choices in both the experiments.

Our findings, that both the current and past environmental richness modulate participants' risk preference, complement previous work showing that people's risk preference is not a stable trait [8,10,12–15,17–20,22]. Utility maximization theories, which are used to explain human risk preference in non-foraging situations, usually assume static risk preference. Though dynamic risk preference can be explicitly modeled into economic utility maximization theories of human risky choice, these theories do not explain peoples' risk preference dynamics. In contrast, ethological theories such as the risk sensitive foraging theory, in which risky choice is guided by maximization of long-term fitness rather than utility, naturally explain dynamic risk preference [34,36–39].

In risk sensitive foraging theory, animals' rate of survival is maximized when animals adjust their risk preference based on the balance between current physiological need and environmental richness [36,37]. This principle has recently been tested in human risky choice experiments, but the experimental paradigms were limited to foraging tasks in which the optimal decision depends on the outcome of past choices or expected future opportunities [43,45,46]. Here we found that participants' risk preference when making risky choice is dynamically modulated even in non-foraging tasks, in which the optimal choice is independent of past outcomes and future opportunities.

The risk-sensitive foraging theory predicts that participants' risk preference will be modulated by the mean rate of gain in the environment as well as the reward accumulated in the past. This suggests that increasing environmental richness should increase risk-aversiveness and vice versa [36,37]. To the best of our knowledge, this prediction has not yet been experimentally investigated in humans. Our first finding, that participants are more risk-averse in the rich than in the poor environment, is consistent with risk sensitive foraging theory's predicted effect of environmental richness on risk preference. The risk sensitive foraging theory further predicts that greater accumulated rewards will increase participants risk-aversiveness and vice-versa. In accordance, we found that participants are more risk-prone after experiencing the poor environment than after experiencing the rich environment.

To investigate whether the concept of accumulated reward in the risk-sensitive foraging formulation explains the effect of past environments on the risk preference in our experiments, we explicitly entered accumulated reward into our regression analysis to explain participants' tendency to gamble. The risk-sensitive foraging theory predicts a negative regression coefficient for accumulated reward (Fig 5; Acc. reward). We found that the effect of accumulated

reward was consistently negative across both experiments, as predicted. In addition, we found that the $\Delta p_p$ regressor (difference in the mean reward probability of the current and past environment), which captures the relative attractiveness of gambling options in the current environment compared to those in the past environments (i.e., $\Delta p_p$ regressor captures how much better the gambling options in the current environment are compared to those in the past environment), was consistently positive and significant in both the experiments. We thus suggest that the effect of the past environment on participants' risk preference was mediated via both the accumulated reward as well as an affect based psychological mechanism.

Taken together, our findings suggest that, like animal foraging behavior, human economic decisions may be more comprehensively explained by the principle of fitness maximization rather than by utility maximization. In addition, our behavioral results support the risk-sensitive foraging theory, which is based on fitness maximization, as a valid model of human decision-making. Lastly, our regression results suggest that affect plays an important role in human economic decisions [15,16,47].

Behavioral economics and recent decision making literature investigate how human decision biases occur by developing computational models that explain observed behavior better than classical theories [6,48–50,23,26,24,30,27]. However, there is no consensus regarding why these biases occur in human economic decision-making [1,51]. To tackle this issue, our primary focus in data analysis centered on descriptive statistics (see Figs 2, 3, 4A and 4B) rather than constructing and testing specific computational models aimed at providing a mechanistic account for the observed shift in risk preference during our experiment. We complemented this approach with a computational model with minimal assumptions. Specifically, we employed the risk-return model to encapsulate participants' risk aversion with a single model parameter within the utility function, and estimated its dynamic transition using a particle filter, which relies solely on the assumption of the Markov property in the temporal transition of the risk aversion parameter (see Fig 4C and 4D). This analytical strategy was deliberately chosen to contrast with the conventional method of analysis, wherein computational models are intricately detailed for the construction of utility functions and its dynamics [23–27]. In doing so, we aimed to provide a unique perspective by adopting an approach which allows us to test the qualitative predictions made by the frameworks of fitness and utility maximization, incorporating minimal assumptions in our computational accounts.

Given the current results, we suggest the following adaptive and mechanistic explanations [52] for the foregoing human economic decision biases. At the adaptive level, we suggest that human economic decisions follow the ecological concepts of fitness maximization such that the same ecological concepts for primary rewards such as food in animals also guide human monetary decisions. At the mechanistic level, we suggest that the foregoing ecological similarity between the treatment of monetary and primary rewards is psychologically implemented via the human affective system in that it treats money in the same way as it treats food or water. In other words, the human affective system has been adaptively tuned for fitness maximization under foraging and uses the same principles when processing economic information. Thus, we suggest that at the emotional level humans treat money similar to food, and the affective inputs to the decision value in the context of monetary decisions automatically follow the principle of fitness maximization.

Fitness maximization as a principle for human economic decision-making is gradually drawing more attention in decision-making literature [40,53,54,41,55]. Fitness maximization prescribes that humans should adaptively adjust their decision tendencies (such as risk preference) as the external environment changes. Thus, to explain human decision-making in dynamic external environments, it is critical to follow the temporal fluctuations in their decision tendencies. Several studies have investigated the effect of dynamic external environments

on the time-dependence of decision parameters [56–59]. Behrens *et al.*, found that the learning rate changes with the volatility of the external environment in a reinforcement learning task [56]. Garrett and Daw found that humans adjust their choices to maximize the rate of gain in response to changes in mean reward rate [57]. Similarly, we here investigated the temporal fluctuations of participants' risk preference. Previous studies compared risk preference between different conditions [13,20,14], or risk preference at different stages of the task [45,46]. However, in the foregoing studies, analysis of risk preference was limited to coarsely discretized time. In our current study, we analyzed trial-by-trial fluctuations in participants risk preference, which is the finest decision timescale in discretized decision tasks.

Our current study has several limitations. First, while the fitness maximization framework correctly predicted the behavior of the participants in our experiments, a critical constraint is our inability to directly measure the participants' internal needs or thresholds which may drive their behavior. However, we note that in animal foraging, it is unlikely that animals adjust their behavior through a direct computation of energy requirements (need); rather, they likely rely on the feeling of hunger (affect). Similarly, we speculate that in our experiments, participants adjusted their choice propensity based on the subjective attractiveness of the option, rather than through a direct computation of need per se. This speculation is partly supported by the results of the regression analysis, but future studies are needed to further elucidate this aspect.

Second, participants were not informed about the environment that they will experience. Thus, they had to learn the reward statistics of the environment to adjust their choice behavior, indicating that participants have robust reward memories. However, we did not assess such learning effects as we refrained from constructing and testing specific computational models with intricate utility functions tailored to specific contexts aimed at providing a mechanistic account for the observed shift in risk preference during our experiment. As such, we did not explicitly test models incorporating a dynamic utility function. Noteworthy concepts in reinforcement learning and decision-making literature, such as the negative contrast effect [60], reference-point centering [26,31,32], range adaptation [26,33,32], and divisive normalization [24,27,29,30], are likely to provide mechanistic accounts to our findings. For example, the two environment-related regressors may be interpretable as range adaptions at two different time scales. The effects of environment-related regressors ($\Delta p_c$ and $\Delta p_p$) and their affect-based interpretation we offered shines light on the question why such adaptation occurs at the two different time scales. We hope that our findings lead to the exploration of computational models with adaptations at multiple time scales in explaining human economic behavior.

Third, related to the previous caveat, our results of model fitting (S5 Fig, S1, S2 and S3 Tables) were inconclusive. The effects of the current and past environments were mediated by different risk parameters depending on the choice of the utility model (the risk-return model, and prospect theory-based multiplicative and additive models) and the choice rule (the sigmoid choice rule and the approach-avoidance choice rule). We also note that the prospect theory-based models might not be entirely suitable for the current experiments due to the constrained range of reward probabilities. Despite this, we hope that the robust observation of the current and past environment effects in our model-free analysis will prompt future studies to delve deeper into the computational underpinnings behind the observed modulations in risk preferences.

Fourth, our experiment exclusively involved gain-only gambles, intentionally omitting the exploration of the impact of losses to avoid the added complexity associated with loss aversion. Future studies may consider introducing sure losses to assess their effects. Fifth, since we did not disclose the total number of trials, participants might have assumed that they lose time or have other opportunity costs for every acceptance. Then, they might rationally reduce

acceptance in the rich environments compared to the poor environments as they might implicitly believe that they could "squeeze" one more trial in during the rich environment if they move on quicker. This effect, somewhat akin to theories on patch leaving, can potentially explain the observed effect of the current environment on participants' gambling behavior.

Sixth, in Experiment 2, the two groups of participants showed a significant difference in the proportion of gambles accepted during the first block, in which both groups experienced the intermediate environment. Thus, direct comparisons of interest for the effect of the current (Fig 2B; $p = 0.026$) and past environments (Fig 3A; $p = 0.125$) were much weaker than anticipated. However, after subtracting the performance during the initial block as baseline performance, the comparisons of interest were robust for both the effects (current environment: S2 Fig, $p < 0.001$; past environment: Fig 3B, $p = 0.006$).

Lastly, the main manuscript reports only group-level analysis, and participant level-analysis, wherever possible, is relegated to the supplementary sections of the manuscript. This is partly due to the high noise associated with the particle-filter estimation of the risk parameter in the state-space model (Fig 4C and 4D), making it impossible to reliably estimate the change in risk preference of a single participant under the current experimental settings. This is most likely because the noise associated with the observation process (the sigmoid choice model) is too high. In addition, the computational cost of the particle filter algorithm was very high, such that a single optimization procedure took about six hours. Additionally, the time-resolved analysis (Figs 4A, 4B and 6) could only be done at the group-level. To maintain consistency, we opted to report only group-level analysis in the main manuscript.

In summary, we have shown that under ecologically valid settings, human economic risk preference depends on both the current and past environmental richness. We propose that future studies should investigate the effect of both current and past environmental factors on people's risk preference when making real economic decisions. In addition, since we found an effect of the past environment in our short duration online experiment, we suggest future studies should investigate the effect of long term exposure to different environments, for example, the effect of economic wealth of one's environment during childhood on human risk preference in adulthood [61].

## Methods

### Ethics statement

This study was approved by Waseda University's Review Board. Informed consent was obtained from all participants prior to their participation in the study.

### Participants

Participants were recruited online via the Japanese crowdsourcing website "Lancers (https://www.lancers.jp/)." For Experiment 1, 300 participants were recruited, and 56 participants were excluded from analysis following the a priori exclusion criteria (see below), leaving a final sample of 244 participants (age: 41.0+/-10.7; 93 female). For Experiment 2, 400 participants were recruited, and 118 participants were excluded from the analysis, leaving a final sample of 282 participants (age: 39.9+/-11.2; 95 female). Participants received a fixed amount of 50 JPY (approximately 0.5 USD at the time of the experiment) upon completing the task. This payment amount aligns with the compensation offered for similar tasks on Lancers. Informed consent was obtained from all participants prior to their participation in the study. Participants were provided with detailed information regarding the study's objectives, procedures, potential risks and benefits, and their rights. This information was presented on a web page, and participants expressed their consent by clicking a radio button with the option "Agree."

The web-based consent form explicitly stated that clicking the "Agree" button indicated the participant's voluntary agreement to participate in the study. A copy of the consent information was made accessible to participants for review, and the date and time of consent were automatically recorded upon submission.

### Behavioral task

Participants first provided their demographic information, such as age and gender. Then task instructions were shown on the screen, which the participants could read at their own pace. Understanding of the task rules was then tested via a brief multiple-choice quiz. Participants were required to answer all the multiple-choice questions correctly to proceed to the next section. If the participants gave a wrong answer, they saw a brief description of the task rules on the screen and were redirected to the task instruction page. Once participants correctly answered all the questions, they proceeded to a practice session consisting of five trials, after which the main experiment started.

In both the practice session and the main experiment, participants repeatedly chose to accept or reject a risky gamble (Fig 1A). Participants were instructed to maximize their gain. The gain associated with a successful gamble (reward magnitude) was explicitly shown at the center of the task screen. The probability of success of the gamble (reward probability) was explicitly shown as the proportion of the pie chart shaded in blue. Accepting the gamble led to the gain of points equal to the reward magnitude if successful and led to nothing otherwise. Rejecting the gamble led to a guaranteed gain of 10 points. The reward magnitude and reward probability were varied trial-by-trial (Fig 1B). Participants indicated their choice by pressing the left or right arrow to select "Yes" or "No". The positions of options were fixed throughout the experiment and randomized between participants. Participants were required to make their choice within three seconds of presentation of the gamble (decision phase). The participants were able to change their choice during the decision phase if they wished to do so. After the three seconds of the decision phase had elapsed, the participants' final choice was highlighted on the screen for one second (confirmation phase). After the confirmation phase, the reward gained by the participants for that trial was displayed at the center of the screen for one second (feedback phase). If the participants failed to respond during the decision phase, no option was highlighted during the confirmation phase, a time-out message was displayed on the screen during the feedback phase, and they did not receive any reward for that trial. Overall, participants who met the inclusion criteria (see below) failed to respond on 0.37% of trials in Experiment 1 and 0.26% of trials in Experiment 2.

Participants experienced multiple blocks of risky choices during the main experiment. Between blocks, we manipulated the distribution of reward probability while keeping the distribution of reward magnitude fixed. Thus, consecutive blocks had different mean reward probabilities, a detail that was not communicated to the participants during the experiment. We refer to the different settings of reward probability distribution as the environment. Each block consisted of one environment only and thus we refer to blocks as environments. In the poor, intermediate, and rich environments, reward probabilities were uniformly distributed with $p = \{0.2, 0.3, 0.4, 0.5, 0.6\}$, $p = \{0.4, 0.5, 0.6\}$, and $p = \{0.4, 0.5, 0.6, 0.7, 0.8\}$, respectively (Fig 1B, $x$-axis). The distribution of reward magnitude was the same for each reward probability and set as $m = \{15, 20, 25, \ldots, 65\}$ (Fig 1B, $y$-axis).

Within each environment (block), catch trials were presented randomly to assess the attention of the participants. Two types of catch trials were presented. One type had a gambling option with a reward magnitude of 10 and a reward probability below 1. Since rejecting the gamble results in a sure gain of 10 points, it is always better to reject the gamble in these trials.

There were eight trials of this type in both the poor and rich environments, and three trials in the intermediate environment. The second type of catch trial had a gambling option with a reward probability of 1 and a reward magnitude above 10. Since accepting the gamble results in a gain greater than 10 points, it is always better to accept the gamble in these trials. There were three trials of this type in each of the poor, rich and intermediate environments. Choice data of participants who responded incorrectly on more than 10% of the catch trials were excluded from the analysis (see exclusion criteria below). The total number of trials during the rich and poor environments was 66 (including 11 catch trials), and that during the intermediate environment was 39 (including 6 catch trials).

We conducted two experiments. In each experiment, two groups of participants experienced different sequences of environment settings. In Experiment 1, one group experienced the poor environment followed by the rich environment (Poor-Rich group or PR group, $n = 108$), and the other group experienced the same environments in the reverse order (Rich-Poor group or RP group, $n = 136$) (Fig 1C, left). There was a self-paced rest between the first and second environments. In Experiment 2, one group experienced the poor environment between two intermediate environments (Intermediate-Poor-Intermediate group or IPI group, $n = 128$), and the other group experienced the rich environment between two intermediate environments (Intermediate-Rich-Intermediate group or IRI group, $n = 154$) (Fig 1C, right). There was no rest period between environments (blocks) in Experiment 2. In both experiments, participants were randomly allocated to either group. All experiments were programmed in JavaScript.

## Exclusion criteria

A priori exclusion criteria were applied to ensure data quality [62]. Specifically, we excluded participants if they: (1) did not finish the task (Experiment 1, $n = 1$; Experiment 2, $n = 23$); or (2) did not respond on at least 5% of the trials ($n = 3$, $n = 7$); or (3) made the same response on all the trials ($n = 1$, $n = 0$); or (4) made an error in at least 10% of the catch trials ($n = 51$, $n = 88$). Overall, 56 participants were excluded from Experiment 1, and 118 participants were excluded from Experiment 2. Consequently, our final sample comprised 244 participants in Experiment 1 and 282 participants in Experiment 2. Note that the data of participants who did not complete the task (i.e., those who met exclusion criterion 1) are not included in source datafiles stored in the GitHub repository provided below.

## Behavioral analysis

To examine the effects of environmental richness on participants' risk preference, we restricted our analysis to only those trials whose gamble parameters, i.e., reward magnitude and reward probability, were experienced in all the environments. Thus, only trials with a reward magnitude greater than 10 and with a reward probability of 0.4, 0.5, or 0.6 were included in the analysis ("Analyzed trials" in Fig 1B). Trials with a reward probability of 0.2 and 0.3 in the poor environment, a reward probability of 0.7 and 0.8 in the rich environment, and catch trials were excluded from the analysis. Trials in which participants failed to respond were also excluded from all analysis except for the analysis using a particle filter (see below), in which they were treated as missing data.

We compared participants' gambling tendency between different environments by comparing the proportion of gambles accepted in each environment

$$q = \frac{number\ of\ gambles\ accepted}{total\ number\ of\ choices}$$

In our main analysis, instead of computing q for each participant (see Supporting information for this measure), we computed $q$ for a group by pooling all the choices of the participants in that group. For example, let the $i^{th}$ group have $N$ participants and let the complete choice data of the $j^{th}$ participant in the $i^{th}$ group be $c_{ij}$. Then, given the pooled choice data for the $i^{th}$ group $\{c_{i1}, c_{i2}, \cdots c_{iN}\}$, we computed the proportion of gambles accepted for the $i^{th}$ group as

$$q_i = \frac{\sum_{j=1}^{N} n_{ij}}{\sum_{j=1}^{N} m_{ij}} \tag{1}$$

where $n_{ij}$ and $m_{ij}$ are the number of times the $j^{th}$ participant in the $i^{th}$ group accepted gambles and made choices, respectively. We tested the statistical significance of the difference in $q_i$ between the two groups in each experiment using the bootstrap procedure (Figs 2B and 3; see below).

In the supplementary analysis, we computed the $q$ for each participant (S3 and S4 Figs). Note that the group-level analysis (main manuscript) and the participant-level analysis (supplementary materials) of the proportion of gambles accepted should return the same results if participants respond on all the trials. Given that participants responded on almost all the trials (failure to respond rates were 0.37% in Experiment 1 and 0.26% in Experiment 2) and to maintain methodological consistency with the analysis of the proportion of gambles accepted across trials and of the risk-aversiveness employing the particle-filter (Fig 4; details provided below and in Discussion), in the main manuscript we chose to present the primary analysis of the proportion of gambles accepted at the group-level (Figs 2B and 3).

## Bootstrap and bootstrap testing

As the proportion of gambles accepted by the $i^{th}$ group $q_i$ represents a single value characterizing the choice tendency of a specific group, $t$-test is not applicable for the between-group comparison. Consequently, to capture the variability in the proportion of gambles accepted (Figs 2B and 3) or other statistics computed for pooled data (Figs 2A and 5), we performed bootstrapping at the participant-level [63].

As mentioned in the foregoing section, we computed $q_i$ for each group (Eq 1). To estimate the variability in $q_i$, we first made a bootstrap sample $\{c_{i1}^*, c_{i2}^*, \ldots c_{iN}^*\}$ of the original pooled choice data $\{c_{i1}, c_{i2}, \cdots c_{iN}\}$, where $c_{ij}^*$ is the $j^{th}$ random draw from the original pooled choice data with replacement. Then the proportion of gambles accepted for this bootstrap sample is computed as

$$q_i^* = \frac{\sum_{j=1}^{J} n_{ij}^*}{\sum_{j=1}^{J} m_{ij}^*} \tag{2}$$

By repeating this procedure $B$ times, we can obtain an empirical distribution of $q_i$ as $\{q_i^1, q_i^2, \ldots q_i^B\}$. We set $B = 9,999$.

To formally test the significance of the test statistics, we performed bootstrap hypothesis testing (Figs 2B and 3). For example, when we tested whether the proportion of gambles accepted for the first group $q_1$ was significantly greater than that of the second group $q_2$, we first computed the test statistic as the difference in the actual proportion of gambles accepted by the two groups

$$D = q_1 - q_2 \tag{3}$$

Then we computed the null distribution of $D$ as

$$D_{null}^b = (q_1^b - q_1) - (q_2^b - q_2) \tag{4}$$

where $q_1^b$ and $q_2^b$ were computed from the foregoing bootstrap procedure as in Eq (2), and $D_{null}^b$ is the $b^{th}$ entry in the empirical distribution of $D_{null}(b = 1,2,\cdots,9999)$. We then compared $D$ with $D_{null}^b$ to compute the $p$-value for the null hypothesis $D \leq 0$ as

$$p = \frac{1 + \sum_{b=1}^{B} I(D_{null}^b - D)}{1 + B} \tag{5}$$

where $I(x)$ is an indicator function which is equal to 1 if $x > 0$, and 0 otherwise.

## Computational models and their fitting

We fitted choices of each participant in each environment with parametric decision models employing various utility models and the choice rules (S5 Fig, S1, S2 and S3 Tables). For the utility models, we used a risk-return model (or also known as mean-variance model) [5], and prospect theory-based multiplicative and additive models. Denoting the reward magnitude and reward probability of an option as $m$ and $p$, respectively, the utility $U$ of an option in the risk-return model is:

$$U = mp - \alpha m^2 p(1 - p) \tag{6}$$

Here the risk-aversion parameter $\alpha$ downregulates the utility of an option according to the associated risk (i.e., variance in the outcome).

For the multiplicative model, the utility of an option is:

$$U = v(m)\pi(p) \tag{7}$$

Here the value function $v(m) = m^\lambda$ includes a risk-proneness parameter $\lambda$, and the Prelec's probability weighting function $\pi(p) = \exp(-(-\log p)^\gamma)$ includes a parameter $\gamma$. The model parameter $\gamma$ is interpretable as a risk-proneness parameter that facilitates the acceptance of gambles. Specifically, for the reward probabilities of the analyzed trials (i.e., $p = \{0.4, 0.5, 0.6\}$), $\pi(p) \geq p$ when $\gamma \geq 1$ and $\pi(p) < p$ when $\gamma < 1$. In this model, we considered combinations of both inclusion and exclusion of the parameters $\lambda$ and $\gamma$ by setting their value to one when excluded. This resulted in consideration of four utility models.

Similarly, for the additive model, the utility of an option is computed as follows:

$$U = (1 - w_p)v(m) + w_p\pi(p) \tag{8}$$

Here the value function $v(m)$ and the probability weighting function $\pi(p)$ are the same as in the case of the multiplicative model, but were normalized and their weighted sum is considered. The model parameter $w_p$ is interpretable in our experimental context as a risk-aversiveness parameter, promoting the participants to reject more gambles. As in the case of the multiplicative model, we considered combinations of both inclusion and exclusion of the parameters $\lambda$ and $\gamma$, resulting in consideration of four utility models.

Choice probability to accept a gamble is commonly determined by a standard sigmoid choice rule:

$$p_{gamble} = \frac{1}{1 + e^{-\beta(U_{gamble} - U_{sure})}} \tag{9}$$

where $U_{gamble}$ and $U_{sure}$ are utilities for the gambling and sure options, respectively. Here the inverse-temperature parameter $\beta$ quantifies choice stochasticity.

Additionally, we considered an approach-avoidance choice rule, where the choice probability to accept a gamble is given as follows [21]:

$$p_{gamble} = \begin{cases} \dfrac{1-\eta}{1+e^{-\beta(U_{gamble}-U_{sure})}} + \eta, \text{if } \eta \geq 0 \\ \dfrac{1+\eta}{1+e^{-\beta(U_{gamble}-U_{sure})}}, \text{otherwise.} \end{cases} \qquad (10)$$

Here the model parameter $\eta$ biases the gambling propensity and is interpreted as a risk-proneness parameter. It facilitates an acceptance of gamble when $\eta > 0$ and vice versa.

These computational models were fitted to the trial-by-trial choices of each group of participants in each environment with a Bayesian hierarchical expectation-maximization method [64]. This method introduces group-level prior distributions, and model parameters for each participant and hyper-parameters of the prior distribution are estimated together. The prior distributions for each parameter are chosen to support appropriate ranges as follows: $\beta \sim$ gamma distribution, $\alpha \sim$ normal distribution, $\gamma \sim$ gamma distribution, $\lambda \sim$ gamma distribution, $w_p \sim$ beta distribution, and $\eta \sim$ beta distribution (transformed to support the range [−1, 1]). The initialization of prior distributions involves setting gamma and normal distributions to support a broad range, while the beta prior distribution is initialized with a uniform distribution.

Since we employed a hierarchical computational model with a group-level prior distribution, we used the integrated Bayesian Information Criterion (iBIC) for model selection, following the recommendation of the previous study [65]. Lower values of iBIC indicate a more favorable model fit.

## Time-resolved analysis of the proportion of gambles accepted

To capture temporal variations in the proportion of gambles accepted across trials, we computed its moving average at each trial (Fig 4A and 4B). Specifically, for the analyzed trials, let the number of participants in a particular group that accepted the gamble on the $i^{th}$ trial be $N_i$, and the number that responded on the $i^{th}$ trial be $M_i$, we computed the moving average of the proportion of gambles accepted for this trial as follows:

$$< q_i > = \frac{\sum_{j=i-k}^{i+k} N_j}{\sum_{j=i-k}^{i+k} M_j} \qquad (11)$$

Here we opted to set $k = 2$, forming a window size of 5 trials (excluding the beginning and end of the experiments, where insufficient preceding or succeeding trials were available).

## State-space model and particle filter

To capture temporal variations in risk preference across trials, we modeled the participants' choices with a state-space model, commonly utilized in time series analysis (Fig 4C and 4D). Our modeling approach involved representing each participant's choice in the $t^{th}$ trial as generated from the reward magnitude and probability encountered by the participant on that trial as well as the risk-averseness $\alpha_t$ specific to the participant's experimental group at that trial. Notably, $\alpha_t$ is a latent variable whose value remains unobservable. Consequently, its temporal dynamics is inferred solely from the participants' choices.

In detail, we express the $i^{th}$ participant's choice in the $t^{th}$ trial, denoted as $y_{ti}$, as a Bernoulli trial,

$$y_{ti} \sim \text{Bernoulli}(\mu_{ti}) \qquad (12)$$

Here, $y_{ti}$ is an indicator variable, that takes 1 if the $i^{th}$ participant accepts the gamble in the $t^{th}$ trial, and 0 otherwise. The probability $\mu_{ti}$ represents the likelihood of the $i^{th}$ participant accepting the gamble in the $t^{th}$ trial, and is modeled as a standard sigmoid choice rule,

$$\mu_{ti} = \frac{1}{1 + e^{-\beta \Delta U_{ti}}} \qquad (13)$$

Here, the inverse temperature $\beta$ captures the stochasticity of the choices and $\Delta U_{ti}$ denotes the excess utility of accepting the gamble over rejecting it. The utility of a choice option was modeled using a risk-return model as follows:

$$\Delta U_{ti} = (m_{ti}p_{ti} - m_s) - \alpha_t(m_{ti}^2 p_{ti}(1 - p_{ti})) \qquad (14)$$

Here, $m_{ti}$ and $p_{ti}$ are the reward magnitude and probability of the choice option that the $i^{th}$ participant encounters in the $t^{th}$ trial, respectively, while $m_S$ is the constant reward magnitude for the sure option throughout the experiment. This utility function ensures that $\alpha_t > 0$, $\alpha_t = 0$, and $\alpha_t < 0$ correspond to risk-averse, risk-neutral, and risk-prone choices tendencies, respectively. We adopted the risk-return model because it demonstrated superior performance among models featuring two parameters—one for risk preference and the other for choice stochasticity. This model provides simple and interpretable insights into the temporal dynamics of risk preference. The choice model specified in the above corresponds to "the observation model" in the terminology of the state-space model.

Furthermore, we posit the stochastic transition of the risk-aversiveness $\alpha_t$ that cannot be observe directly as follows:

$$\alpha_t \sim \text{Cauchy}(\alpha_{t-1}, \sigma) \qquad (15)$$

Here, $Cauchy(x_0, \sigma)$ is the Cauchy distribution with density $p(x) = \sigma/(\pi(\sigma^2 + (x - x_0)^2))$. The free parameter $\sigma$ captures the flexibility in the stochastic transitions of $\alpha_t$, and larger value of $\sigma$ allows more flexible transitions. This specification of the stochastic transition of a latent variable is referred to as "the system model" in the terminology of the state-space model.

With the aforementioned observation model and system model, we employed a particle filter algorithm [66] to estimate the temporal dynamics of risk-aversiveness $\alpha_t$ that cannot be observed directly. This hidden variable was inferred solely from participants' pooled choices within each experimental group.

The foregoing model contains two free parameters: the flexibility $\sigma$ in the transitions of risk-aversiveness and the inverse temperature $\beta$. The values for these parameters were optimized so that the likelihood for pooled choices was maximized for each group. Specifically, we performed a grid search over the parameter grid of fifty equally spaced points between $10^{-5} \leq \log\sigma \leq 10^{-1}$ and $10^{-1} \leq \beta \leq 1$. In each iteration, the particle filter algorithm was run with $N = 10,000$ particles, and with the initial estimate of risk-aversiveness $\alpha_0 \sim Norm(0,1)$. The smoothing was carried out by the fixed-lag smoothing algorithm with a lag $L = 20$ as recommended by Kitagawa[67]. We took the median of the smoothing distribution to be the optimal estimate of the risk-aversiveness $\alpha_t$. For unanalyzed trials (see Fig 1) and trials with no response, the participants' choices were treated as missing data (i.e., for all environments, we only used those trials which had a reward probability $p = \{0.4, 0.5, 0.6\}$ and on which participants responded, as in the rest of the analysis).

We validated the particle filter's estimation by applying it to simulated choices (S6 Fig). Choices were simulated assuming a step change in the risk-aversiveness $\alpha$. The particle filter was able to recover the sudden change in risk-aversiveness when simulation parameters were set close to that of the real experiment (number of trials in a block = 66, number of participants = 120, and $\Delta\alpha = 0.2 \times 10^{-2}$), and Cauchy noise was assumed in the system model.

## Logistic regression and simulation

To unravel the variables responsible for the pattern of shifts in the participants' risk preference observed in our experiments, we analyzed participants' choices with a logistic regression at the group level (Fig 5). Denoting the reward magnitude of a given gamble as $m$ and reward probability as $p$, we used the following regressors in our analysis: i) the expected value (EV = $mp$), ii) the risk of the gambling option (Risk = $m^2 p(1-p)$), iii) the deviation of reward probability of the current option from the mean reward probability of all gambling options in the environment ($\Delta p_c = p - p_{c_{mean}}$, where $p_{c_{mean}}$ represents the mean reward probability in the current environment), iv) the difference between the mean reward probability in the current and past environments ($\Delta p_p = p_{c_{mean}} - p_{p_{mean}}$, where $p_{p_{mean}}$ represents the mean reward probability in the previous environment), v) the previous choice (Prev. choice, an indicator variable that takes 1 if the previous choice was to accept the gamble, and 0 otherwise), vi) the previous reward (Prev. reward, number of points gained in the previous trial), vii) the outcome of the previous gamble (Prev. success, an indicator variable that takes 1 if the previous choice was to accept the gamble and the gamble was successful, and 0 otherwise), and viii) the accumulated reward (Acc. reward, accumulated number of the point up to the current trial normalized at the participant level, normalize at participant-level). All regressors were computed for every trial, pooled for each experiment, and subsequently z-scored. The model fitting process was then carried out exclusively with the analyzed trials depicted in Fig 1B. The significance of the regression coefficients was tested by the bootstrap test described above (See Methods: Bootstrap and bootstrap testing).

To test the importance of environment-related regressors $\Delta p_c$ and $\Delta p_p$ in explaining participants' risk preference dynamics, we simulated the choices of 120 participants with the estimated values of regression coefficients (Fig 6) and computed the moving average of the proportion of the gambles accepted. A regressor whose z-scored values were dependent on the history of choices and/or rewards was pseudo-standardized by the mean and standard deviation computed for the corresponding experimental group. First, we validated our simulations by showing that simulations which included all the regressors whose regression coefficients were consistently significant across the two experiments qualitatively reproduced the temporal dynamics of the participants' risk preference seen in the real data (simulation 1; Fig 6A). Next, we tested whether simulations with all the foregoing regressors except the environment-related regressors $\Delta p_c$ and $\Delta p_p$ could reproduce the temporal dynamics of the participants' risk preference seen in the real data (simulation 2; Fig 6B). For this simulation, we replicated the procedure of the simulation 1, except that the regression coefficients estimated for the model excluding $\Delta p_c$ and $\Delta p_p$ were employed in simulating choices.

## Supporting information

**S1 Table. Model comparisons results.** Integrated Bayesian information criterion (iBIC) of models with the sigmoid and approach-avoidance choice rules were computed and compared. A lower iBIC value indicates a better model performance.
(PDF)

**S2 Table. Results of hypothesis testing for the estimated parameters of the models with the sigmoid choice rule.**
(PDF)

**S3 Table. Results of hypothesis testing for the estimated parameters of the models with the approach-avoidance choice rule.**
(PDF)

**S4 Table. Correlation structure and VIF of regressors entered into the main regression analysis.** ***$p < 0.001$ (Bonferroni corrected).
(PDF)

**S1 Fig. Participants inclination toward gambling exhibited a significant increase as the reward magnitude and reward probability increased.** Proportions of gambles accepted for various combinations of gambling parameters were computed across all participants in each experiment, and sigmoid curves were fitted using binomial regression. (A) Binomial regression results with reward magnitude as the explanatory variable showed a significantly positive slope for the sigmoid curve in both Experiment 1 (Left, slope = 0.044, 95% CI = [0.040, 0.049]) and Experiment 2 (Right, slope = 0.041, bootstrapped 95% CI = [0.038, 0.045]). (B) Similarly, binomial regression results with reward probability as the explanatory variable demonstrated a significantly positive slope for the sigmoid curve in both Experiment 1 (Left, slope = 12.5, bootstrapped 95% CI = [11.4, 13.6]) and Experiment 2 (Right, slope = 12.1, 95% CI = [11.3, 13.1]).
(TIF)

**S2 Fig. IRI group exhibited a significantly greater decrease in proportion of gambles accepted compared to the IRI group in the second block.** A significance test for the effect of environmental richness on gambling propensity. Change in the proportion of gambles accepted from the first to the second environment are computed for each group and compared (bootstrap test, one-tailed). Note that in the first environment both groups experienced the intermediate environment, and in the second block IPI and IRI group experienced the poor and rich environments, respectively. The shaded regions represent the variation, due to random samplings of participants, in the mean change in the proportion of gambles accepted as computed by bootstrap. The colored and black horizontal lines correspond to the mean and 95% confidence intervals, respectively. ***$p < 0.001$.
(TIF)

**S3 Fig. Participants accepted significantly more gambles in the poor environment than in the rich environment.** Mean proportion of gambles accepted by each group during the first environment of Experiment 1 (left) and during the first and second environments of Experiment 2 (right). In both Experiment 1 and Experiment 2 (second environment), participants accepted significantly more gambles during the poor environment than during the rich environment. In contrast, when both groups (IPI and IRI) were experiencing the same intermediate environment in Experiment 2 (first environment), the difference in the proportion of gambles accepted was not significant. One-tale bootstrap tests were used for the test of significance. The circles and error bars represent the mean and standard error. The bubble plots represent the distribution of the proportion of gambles accepted by each participant. *$p < 0.05$; ***$p < 0.001$.
(TIF)

**S4 Fig. Participants accepted significantly more gambles after experiencing the poor environment than after experiencing the rich environment.** (**A**) In Experiment 1, the group that

experienced the poor environment first (PR group) accepted significantly more gambles than the group that experienced the rich environment first (RP group) across the whole experiment (left), and in both the poor (middle) and rich (right) environments. (**B**) In Experiment 2, the IPI group accepted more gambles than the IRI group in the final block (second intermediate environment), although this trend did not reach significance. One-tale bootstrap tests were used for the test of significance. The circles and error bars represent the mean and standard error. The bubble plots represent the distribution of the proportion of gambles accepted by each participant. $^*p < 0.05$; $^{**}p < 0.01$; $^{***}p < 0.001$.
(TIF)

**S5 Fig. Estimated parameters of the risk-return and prospect theory-based models with the sigmoid choice rule.** Model parameters were estimated from the trial-by-trial choices of each participant for each environment. (A) Risk-aversiveness $\alpha$ of the risk return model estimated for Experiment 1. (B) Risk-proneness $\gamma$ of probability weighting function (left) and risk-proneness $\lambda$ of the value function (right) of the multiplicative model estimated for Experiment 1. (C) Risk-proneness $\gamma$ of probability weighting function (left), risk-proneness $\lambda$ of the value function (middle), and risk-aversiveness $w_p$ (right) of the additive model estimated for Experiment 1. (D-F) Mirroring the format of A-C but are plotted for Experiment 2. The filled circles and colored bars denote the means and their respective standard errors. Unfilled circles signify the estimated values of the parameters for each participant within each environment.
(TIF)

**S6 Fig. Validation of the particle filter.** Choices were simulated assuming a step change in the risk-aversiveness $\alpha$ (black). Then, the particle filter was used to estimate the risk-aversiveness from the simulated choices (green and purple lines). The particle filter was able to recover the sudden change in risk-aversiveness when simulation parameters were set close to that of the real experiment (number of trials in a block = 66, number of participants in a group = 120, and $\Delta\alpha = 0.2\times10^{-2}$) and Cauchy noise was assumed in the system model (green line, top left). The shaded regions represent the 90% credible intervals of the risk-aversiveness estimate by the particle filter.
(TIF)

**S7 Fig. Dynamics in the simulated choices generated from the risk-aversiveness estimated by the particle filter.** (**A**) A moving average of the proportion of gambles accepted is computed for the simulated choices of each group ($n = 120$) across Experiment 1 (left) and Experiment 2 (right). Each data point represents the proportion of gambles accepted in current trial and the preceding and succeeding two trials, forming a window size of 5 trials. Error bars represent standard errors. (**B**) Mean proportion of gambles accepted in each environment by each group. In both experiments, simulation showed that the proportion of the gambles accepted is significantly higher during the rich as compared to the poor environment ($p < 0.001$ for Experiment 1 and 2; top). In addition, the proportion of the gambles accepted was significantly higher after experiencing the poor rather than the rich environment ($p < 0.001$ for poor & rich environment in Experiment 1 and for the second intermediate environment in Experiment 2; bottom). One-tailed unpaired t-tests with Bonferroni correction were used for statistical comparisons of proportion of gambles accepted (except for the comparison for the first intermediate in Experiment 2 where two-tailed test was employed given the initial prediction that there is no difference.) $^{***}p < 0.001$.
(TIF)

**S8 Fig. Stability of the estimated regression coefficient.** Regression coefficients estimated for each experiment with the full model (white; identical to Fig 5), full model with L1

regularization (blue), the model with only the primary regressors (EV, Risk, $\Delta p_c$, and $\Delta p_p$; pink), and the model excluding regressors with a VIF greater than five in either experiment (grey; see S4 Table for the values of VIF). Error bars represent 95% confidence bounds computed from bootstrap.
(TIF)

**S9 Fig. Regression coefficient estimated for the model excluding the environment-related regressors.** Mirroring the format of Fig 5, but plotted for the model excluding the two environment-related regressors, $\Delta p_c$ and $\Delta p_p$. Regressors with consistently significant regression coefficients across the two experiments are noted in bold. Error bars represent 95% confidence bounds computed from bootstrap. Two-tailed one-sample bootstrap tests against zero with Bonferroni correction were used for statistical tests of regression coefficients. $^{**}p < 0.01$.
(TIF)

**S10 Fig. $\Delta p_c$ and $\Delta p_p$ are necessary for recovering the current and past environment effects, respectively.** (**A, B**) The simulation results of the model excluding $\Delta p_c$ ($n = 120$ for each group). The moving average of the proportion of gambles accepted for each trial (A) and their environment-wise average (B) were calculated for each group. (**C, D**) Mirroring the format of A & B but plotted for the model excluding $\Delta p_p$. (**E, F**) Mirroring the format of A & B but plotted for the model excluding Prev. choice. (**G, H**) Mirroring the format of A & B but plotted for the model excluding Acc. reward. (**A, C, E, G**) Error bars represent standard errors. Shaded regions are 95% confidence intervals of the proportion of gambles accepted computed for the real data. (**B, D, F, H**) Horizontal bars and unfilled dots represent the mean and raw values of the proportion of gambles accepted, respectively. One-tailed unpaired t-tests with Bonferroni correction were used for statistical comparisons of the mean proportion of gambles accepted (except for the comparison for the first intermediate in Experiment 2 where two-tailed test was employed given the initial prediction that there is no difference.) $^*p < 0.05$; $^{***}p < 0.001$.
(TIF)

**S11 Fig. Incorporating regressors for the choices made two and three trials ago did not alter the outcome.** The simulation results of the model including regressors for two and three trials ago on top of the regressors in the original model ($n = 120$ for each group). (**A**) The moving average of the proportion of gambles accepted for each trial. Error bars represent standard errors. Shaded regions are 95% confidence intervals of the proportion of gambles accepted computed for the real data. (**B**) The mean proportion of the gamble accepted for each environment. Horizontal bars and unfilled dots represent the mean and raw values of the proportion of gambles accepted, respectively. One-tailed unpaired t-tests with Bonferroni correction were used for statistical comparisons of the mean proportion of gambles accepted (except for the comparison for the first intermediate in Experiment 2 where two-tailed test was employed given the initial prediction that there is no difference.) $^{***}p < 0.001$.
(TIF)

## Author Contributions

**Conceptualization:** Yasuhiro Mochizuki, Norihiro Harasawa, Mayank Aggarwal, Chong Chen, Haruaki Fukuda.

**Data curation:** Yasuhiro Mochizuki, Norihiro Harasawa.

**Formal analysis:** Yasuhiro Mochizuki.

**Investigation:** Yasuhiro Mochizuki, Norihiro Harasawa.

**Writing – original draft:** Yasuhiro Mochizuki, Mayank Aggarwal.

**Writing – review & editing:** Yasuhiro Mochizuki, Norihiro Harasawa, Mayank Aggarwal, Chong Chen, Haruaki Fukuda.

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
