## [Decision Letter · Decision Letter 0]

3 Aug 2023

Dear Dr Mochizuki,

Thank you very much for submitting your manuscript "Foraging in a non-foraging task: fitness maximization explains human risk preference dynamics under changing environment" for consideration at PLOS Computational Biology.

As with all papers reviewed by the journal, your manuscript was reviewed by members of the editorial board and by several independent reviewers. In light of the reviews (below this email), we would like to invite the resubmission of a significantly-revised version that takes into account the reviewers' comments.

We cannot make any decision about publication until we have seen the revised manuscript and your response to the reviewers' comments. Your revised manuscript is also likely to be sent to reviewers for further evaluation.

Sincerely,

Stefano Palminteri

Academic Editor

PLOS Computational Biology

Daniele Marinazzo

Section Editor

PLOS Computational Biology

Reviewer's Responses to Questions

**Comments to the Authors:**

Reviewer #1: This manuscript is about risky decision making about accept-reject reward gambles, combining it with an interesting concept from ecology, i.e. dynamic contextual changes in risk sensitivity (primarily described in the risk sensitive foraging literature, but more recently applied to human behaviour and neuroimaging). What is particularly novel about this study, compared to earlier work, is that it takes such an ecological view explicitly towards “non-foraging” type gambles, i.e. gambles without actual sequential dependencies or homeostatic needs. The authors show convincingly that regardless of this, something is clearly going on with participants risk taking, that cannot be explained simply through the notion of participants making independent trial by trial gambling decisions without contextual dynamics playing a part.

Overall, this is a great idea, well executed and with convincing data.

However, I think the manuscript could nonetheless benefit from some more clarifying analyses on the nature of their effects, some clarifications and some more contextualization in regards to the literature. Below a point-by-point breakdown of my comments:

Major:

1) I very much liked their idea of bringing ecological theories into decision sciences. However, at times I wasn’t quite sure how everything fit together. Normally risk sensitive foraging theory is primarily about homeostasis and setpoints. Even in their example of starving animals its about meeting a minimal amount to ensure survival that pressures animals to take risks. However, in their study there are no equivalent analyses that would suggest a specific setpoint. Of course, such analysis would be difficult to conduct as those thresholds would have to be bootstrapped somehow from the data and might differ between people, but that difference in the concept behind their inspiration and their approach should be noted. However, it is probably true that if participants have a reasonably high threshold for necessary earnings that this would push them towards increased risk taking in poor environments (and that this effect could persist if they don’t completely update their expectations about future gambles) and this could be potentially shown in a simulation but so far the authors do not explicitly show this link, making their results less about homeostatic decision making and more about general contextual risk modulation.

2) I wasn’t quite sure about the correlational structure of some of the parameters entered into the regression analysis. It would be very useful to see a correlation matrix of all multiple regression analysis. Specifically, I was wondering whether some regressors such as previous reward, but more importantly accumulated reward aren’t quite correlated with the environment type? I am somewhat worried that the authors inadvertently omitted the nonmatched trials outcomes when computing the accumulated rewards, removing such correlations but distorting the meaning of accumulated rewards as it now does not reflect the actual reward experience of the participants anymore. Additionally, related to point 1, accumulated reward related effects could be dynamic and based on where the agent is relative to their goals. In other words, testing the linear impact of accumulated reward might not be the right thing to do, but instead a binning of risk aversion or decision parameters by accum R bins to see whether there is a “peak”?

3) I couldn’t tell from the analysis, whether participants were generally risk averse or risk seeking and whether they came closer to the ideal risk proneness in the rich or poor environment as defined by optimal i.e. nonbiased decision making? Additionally, I couldn’t find any comparisons of decision models that take different distortions into account such as KT-models or linear weighting of probability and magnitude rather than multiplication of the objective values. A linear weight analysis might be particularly interesting because it could test whether changes in context specifically impacts the use of magnitude or probability in risky decision making (on top of effects on their general risk taking bias).

4) There is other literatures that could explain at least some of their findings. For example, RL has long had the idea of negative contrast effects, i.e. that learning happens in comparison to the current average reward rate. Generalized, such an effect could explain why participants accept more gambles in a poor environment, without resorting to foraging style language, as negative contrast effects can be beneficial in learning to identify good items in bad environments.

Also, If people assume they lose time or have other opportunity cost for every acceptance, they might rationally reduce acceptance in rich environments compared to poor ones as they might implicitly believe that could “squeeze” one ore trial in in rich environments if they move on quicker (however, some of this reasoning is of course related to patch leaving theories).

These concepts need to be discussed!

5) Lastly, I was a bit surprised that the effect when modelled as risk aversion or in a regression was so much larger compared to when calculated through propoertion of gambles accepted. In other words, why is the dynamic risk aversivness plus particle filter more powerful? This is not to question their results, I just lack intiuition on the source of this difference and likely the reader will as well.

6) You stated “Participants received 50 JPY (approximately 0.4 USD) upon completion of the task. “ This seems very low as payment or am I misunderstanding this and it isn’t the overall payment? Also were participants paid on points earned or not? This is important information for a gambling study (I apologies if it was mentioned somewhere but I missed it).

Minor:

- Risk could be variance (highest at 50/50) or odds of not winning (highest at lowest probability) so specify earlier clearly which definition you are using (economists and psychologist use the word risk to mean different things and this can be confusing when not clearly stated which one is used).

- It is called “Risky choices” in english not “risk choices”

- Why did you not analyze the lowest mag (10) that overlapped between environments or is this a ploting mistake?

- I didn’t understand why the simulation without the effect would show the opposite trend? Please clarify

- It wasn’t always clear why bootstrapping analyses were used (I understand it for the particle filter but some rather conventional measures also had it without explanations e.g. “Choices shown in A were pooled for each environment and the proportion of gamble accepted were compared between environments (bootstrap test, one-tailed). The shaded regions represent the variation, due to random samplings of participants, in the mean of the proportion of gambles accepted as computed by bootstrap. The colored and black horizontal lines correspond to mean and 95% confidence intervals of the proportion of gambles accepted, respectively. *p < 0.05; ***p < 0.001.”

- Fig 2. Could be clearer as it took me a long time to understand what was being plotted. Maybe have separate legends for subpanels but make them clearer?

- It would be good for readers if additional explanation of the rationale behind Experiment 2 happened earlier and was more thorough.

- In Fig 3 A make y axis all the same so that the data points can be compared across subpanels.

- You stated “(B) In experiment 2, the IPI group accepted more gambles in the second intermediate environment than the IRI group, although this trend did not reach significance (bootstrap test, one-tailed). “ but if it isn’t significant I wouldn’t use this language just because one is numerically higher than the other, as people normally assume statistical significance if you say “x is higher than y”. you can say “while x was numerically higher than y, it was not statistically significantly so”

- I was surprised how big the previous choice effect was. It suggests to me that the stickiness does not end there, but t-2 and t-3 probably also have an impact (although unlikely this would change any other findings).

- Its Experiment, not Experient. Saw this typo a couple of times.

- Why is the change in risk aversion parameter in Exp2 so sudden in the middle bit but more smooth in Exp 1 in the simulation in Fig 5?

- Why sure gains of 10 rather than probability of losing a certain amount? Not that this is wrong, but it would be nice to see an argument for it in the paper.

- Their order effect you show can only be obtained if the participants have a good reward memory, which they appear to have. Might be useful to point that out.

Reviewer #2: The review is uploaded as an attachment.

Reviewer #3: In this article, Mochizuki et al. show in two experiments that decisions under risk are influenced by the local context that they manipulated by changing the reward probability distribution. More specifically, participants were more risk averse during and after the rich environment compared to the poor environment.

I found the article well written and the experimental design neat. Yet, I have some concerns and suggestions which I hope will help improve the paper.

At the conceptual level:

1) I am not quite sure why the authors framed their study with a distal mechanism (fitness maximisation as per foraging decision theory) to explain that the local context influences decisions. It is typically difficult to measure the equivalent in money to “the current physiological need” that would match the “environmental richness”. I found the discussion lines 485-500 rather indirect. Other proximal theories like range adaptation or change in the (prospect theory utility function) reference point would also explain the results in a more direct way. For example, a reward probability of 0.6 is the best in the poor environment and may have the maximal value (as per range adaptation) or a positive utility (with a dynamical reference point) whereas it would have a low value or a negative utility in the rich environment and should be chosen less. This theoretical framework seems more in line with the regression analyses the authors performed (delta pp and delta pc).

Regarding the design:

2) It seemed to me that participants were not instructed which environment they were in, which was confirmed in the last paragraphs of the discussion. I suggest to state it in Figure 1 caption and in the result section as well as in the methods.

Regarding the model-agnostic results:

3) Is the decrease in gambling following the reach environment constant across the whole-session? I suggest the authors to show the model-agnostic temporal dynamics of gambling (potentially smoothed over a few trials). This would help see whether the change in risk attitude is immediate (unexpected) or progressive -and if so, the transition slope. This may be especially relevant to unpack figure 3B: the observed trend may result from a significant difference early in the last intermediate environment but not at the end due to adaption. I am aware of the attempt to model such dynamics, but model-agnostic data would be much more convincing.

4) In Experiment 2, I believe that the proportion of gambles accepted in the second environment could be compared to that of the first environment (i.e. rich 2 – intermediate versus poor 2 – intermediate). This sounds like a fair comparison and may help strengthen the otherwise edgy inference (p = 0.026 one tailed). This also applies to figure 3B where the results are reported as a trend.

5) Some key p-values are one-tailed and yet on the edge (e.g. figure 2B, p = 0.026). This may be improved following the previous suggestions. If not, I guess that it should be made explicit in the study limitations. Similarly, note that a trend for a p-value equals to 0.125 one-tailed is also a bit edgy (line 255).

6) Still related to the statistics, I was surprised by the use of a fixed effect analysis (ignoring participants individual variance). Why not only using a more traditional random-effect for all the analyses a sin Figure S1 & 2 for all the data? The authors mentioned in the discussion that data were too noisy to do so. Could they elaborate a bit more? Do they think that adding more trials, blocks, and participants would help? Why not then using mixed-effect analyses, particularly well-suited for data with low power?

Regarding the model-based analyses:

7) The current particle filter model lacks of goodness of fit metrics (e.g. pseudo-r2 and balanced accuracy) and validation by e.g. overlapping the model predictions with the actual data. It is rather unclear whether the changes in alpha presented in figure 3 C & D are, if even impactful, to the same magnitude order as in the choice data.

8) Was this model used on all trials or only the intermediate ones?

9) Risk averseness simulations suggest that there is no difference in Experiment 1 between rich and poor environment in the second block. Yet, it seems not to be the case looking at Figure 3A (i.e. there seems to be a significant difference). Can the authors report 

---

## [Decision Letter · Decision Letter 1]

7 Mar 2024

Dear Dr Mochizuki,

As you can see we are going to accept the paper, however we will kindly ask to fix the minor concerns raised by R2. 

Thank you very much for submitting your manuscript "Foraging in a non-foraging task: fitness maximization explains human risk preference dynamics under changing environment" for consideration at PLOS Computational Biology. As with all papers reviewed by the journal, your manuscript was reviewed by members of the editorial board and by several independent reviewers. The reviewers appreciated the attention to an important topic. Based on the reviews, we are likely to accept this manuscript for publication, providing that you modify the manuscript according to the review recommendations.

Sincerely,

Stefano Palminteri

Academic Editor

PLOS Computational Biology

Daniele Marinazzo

Section Editor

PLOS Computational Biology

Reviewer's Responses to Questions

**Comments to the Authors:**

Reviewer #1: The authors addressed all my points. Congratulations on an interesting study!

Reviewer #2: I appreciate the authors' efforts in addressing my previous concerns. I believe they have made satisfactory adjustments that have greatly improved the manuscript.

However, I would like to provide some minor suggestions and noticed a few typos that could be corrected:

1. In Figure 1D, it seems to me that the circles represent the optimal risk preference, but this information is not mentioned in the legend. Adding this information would be helpful.

2. L.216: In this section, the authors mention "participants' needs" as the minimum number of points within a set of trials required for survival (Fig 1D). To enhance clarity, it would be helpful if the authors explicitly specify the number of points considered for high, moderate, and low needs.

3. L.377: The statement mentions, “The predicted changes in risk-preference parameters were significant for the risk-aversiveness of the risk-return model, and the risk-proneness of the additive model, consistently across Experiment 1 and Experiment 2.” However, based on my interpretation of S7 Table, the risk-proneness of the additive model is not consistently significant across both experiments. Clarification on this point is needed.

4. L.389: “The model comparison of the approach-avoidance version of the models with iBIC showed that the best model was the additive model including both and (S8 Table)”. It seems to me that there may be a reference error and it should be S5 Table instead.

5. Model Choice in Particulate Filter: In the results section, it would be helpful if the authors explained why they continue to use a risk-return model in the particulate filter when the additive model consistently outperforms it in the previous section.

6. L.550: I believe there is a typo in the term “Acc. Reared” and it should be corrected to “Acc. Reward”.

7. It would be helpful if the authors provided an explanation of the rationale for using iBIC for fitting computational models and BIC for regression analysis. Clarifying the differences between the two methods would be appreciated.

8. L.639: The statement mentions, “Next, we simulated participants' choices with significant regressors common to both experiments (shown in bold in Fig 5).” However, it appears that ∆ and ∆ are not bold in Fig. 5 as indicated.

**Have the authors made all data and (if applicable) computational code underlying the findings in their manuscript fully available?**

Reviewer #1: **No: **I checked that two files had been deposited on Github for the data but couldn't find a statement regarding the analysis code itself. I am not sure PLOS requires this but just wanted to flag this, in case.

Reviewer #2: None

PLOS authors have the option to publish the peer review history of their article (what does this mean?). If published, this will include your full peer review and any attached files.

Reviewer #1: No

Reviewer #2: No

Figure Files:

Data Requirements:

Reproducibility:

References:

---

## [Editor Report · Decision Letter 2]

16 Apr 2024

Dear Prof. Mochizuki

We are pleased to inform you that your manuscript 'Foraging in a non-foraging task: fitness maximization explains human risk preference dynamics under changing environment' has been provisionally accepted for publication in PLOS Computational Biology.

Best regards,

Stefano Palminteri

Academic Editor

PLOS Computational Biology

Daniele Marinazzo

Section Editor

PLOS Computational Biology

---

## [Editor Report · Acceptance letter]

3 May 2024

PCOMPBIOL-D-23-00980R2 

Foraging in a non-foraging task: Fitness maximization explains human risk preference dynamics under changing environment

Dear Dr Mochizuki,

I am pleased to inform you that your manuscript has been formally accepted for publication in PLOS Computational Biology. Your manuscript is now with our production department and you will be notified of the publication date in due course.

With kind regards,

Zsofia Freund
